# CS-DIALOGUE: A 104-HOUR DATASET OF SPONTANEOUS MANDARIN-ENGLISH CODE-SWITCHING DIALOGUES FOR SPEECH RECOGNITION

## ABSTRACT

Code-switching (CS), the alternation between two or more languages within a single conversation, presents significant challenges for automatic speech recognition (ASR) systems. Existing Mandarin-English code-switching datasets often suffer from limitations in size, spontaneity, and the lack of full-length dialogue recordings with transcriptions, hindering the development of robust ASR models for real-world conversational scenarios. This paper introduces CS-Dialogue, a novel large-scale Mandarin-English code-switching speech dataset comprising 104 hours of spontaneous conversations from 200 speakers. Unlike previous datasets, CS-Dialogue provides full-length dialogue recordings with complete transcriptions, capturing naturalistic code-switching patterns in continuous speech. We describe the data collection and annotation processes, present detailed statistics of the dataset, and establish benchmark ASR performance using state-of-the-art models. Our experiments, using Transformer, Conformer, and Branchformer, demonstrate the challenges of code-switching ASR, and show that existing pre-trained models such as Whisper still have the space to improve. The CS-Dialogue dataset will be made freely available for all academic purposes.

## 1 INTRODUCTION

Code-switching (CS) refers to the practice of alternating between two or more languages within a single conversation or utterance (Moyer, 2002). It is a common linguistic phenomenon in multilingual communities and occurs in various communication settings, including spoken dialogues, social media, and written texts. The increasing prevalence of code-switching presents significant challenges for automatic speech recognition (ASR) systems, as they must effectively handle complex acoustic and linguistic variations across different languages (Yılmaz et al., 2018).

Traditional ASR systems, predominantly trained on monolingual data, struggle with code-switched speech due to mismatches in phonetic inventories, syntactic structures, and language switching patterns (Mustafa et al., 2022; Zhou et al., 2024). These challenges necessitate the development of specialized ASR models and high-quality datasets tailored for code-switching scenarios. Despite recent advancements, existing Mandarin-English code-switching speech corpora remain limited in size, spontaneity, and accessibility, restricting further research and model development.

Table 1 provides an overview of publicly available Mandarin-English code-switching datasets. Many existing corpora (Shen et al., 2011; Wang et al., 2016; Li et al., 2022) focus on read speech or constrained domains, lack full transcriptions or are not publicly accessible. Crucially, most datasets comprise isolated code-switching utterances rather than full dialogues, limiting their utility for studying naturalistic speech patterns and contextual dependencies (Chang et al., 2023).

To address these gaps, we introduce **CS-Dialogue**, a novel large-scale Mandarin-English code-switching speech dataset consisting of 104 hours of spontaneous conversations from 200 speakers. Unlike prior work, our dataset provides full-length dialogue recordings with complete transcriptions, capturing naturalistic code-switching phenomena in continuous speech. This dataset enables more comprehensive investigations into code-switching ASR beyond isolated utterances. CS-Dialogue is, to the best of our knowledge, the largest publicly available dataset of spontaneous Mandarin-English code-switching dialogues with full transcriptions.

Table 1: Comparison of Mandarin-English code-switching speech datasets. "Tr." indicates whether transcripts are available, "Avail." specifies whether the dataset is publicly accessible, and "Full-dialogue" denotes whether full-length dialogue recordings and transcriptions are provided.

| Dataset | Duration (h) | #Speakers | Audio Type | Tr. | Avail. | Full-dialogue |
|---|---|---|---|---|---|---|
| CECOS (Shen et al., 2011) | 12.1 | 77 | Read | No | No | No |
| OC16-CE80 (Wang et al., 2016) | 80 | 1400+ | Read | Yes | No | No |
| ASRU (Shi et al., 2020) | 240 | N/A | N/A | Yes | No | No |
| TALCS (Li et al., 2022) | 587 | 100+ | Online Teaching | Yes | Yes | No |
| DOTA-ME-CS (Li et al., 2025) | 18.54 | 34 | Read | Yes | Yes | No |
| SEAME (Lyu et al., 2010) | 30 | 157 | Conversation | Yes | Paid | No |
| Li et al. (Li et al., 2012) | 36 | N/A | Conversation | Partial | No | No |
| ASCEND (Lovenia et al., 2022) | 10.62 | 23 | Conversation | Yes | Yes | No |
| Ours | 104.02 | 200 | Conversation | Yes | Yes | Yes |

In this paper, we describe the data collection and annotation processes, present key characteristics of the dataset, and evaluate its impact on ASR performance through baseline experiments. Our contributions are summarized as follows:

- We construct a large-scale, spontaneous Mandarin-English code-switching speech corpus with full-length dialogue transcriptions, filling the gap of publicly available datasets in this domain.

- We detail the data collection and annotation processes, ensuring high transcription accuracy and providing a well-documented resource for future research.

- We establish benchmark ASR performance on our dataset using state-of-the-art models, offering insights into the challenges of code-switching ASR.

## 2 RELATED WORK

Existing Mandarin-English CS speech datasets can be broadly categorized into read speech and spontaneous speech corpora. Read speech datasets typically contain pre-defined sentences that participants are instructed to read aloud, offering controlled phonetic and linguistic variations but lacking the spontaneity of natural conversations.

The CECOS dataset (Shen et al., 2011) is one of the earliest Mandarin-English CS corpora, comprising 12.1 hours of read speech from 77 speakers at National Cheng Kung University in Taiwan. While it includes code-switching utterances, it lacks publicly available transcriptions. OC16-CE80 (Wang et al., 2016) significantly expands the scale, offering 80 hours of read speech from over 1400 speakers, with transcriptions available but not open-sourced. The ASRU dataset (Shi et al., 2020), developed for an ASR challenge, contains 240 hours of predominantly Mandarin speech interspersed with some English. Although transcriptions exist, the dataset is not publicly accessible.

More recent datasets, such as DOTA-ME-CS (Li et al., 2025), offer open-source transcriptions and introduce AI-based augmentation techniques (e.g., timbre synthesis, speed variation, and noise addition) to enhance diversity. However, its scale remains relatively small, with only 18.54 hours from 34 speakers. TALCS (Li et al., 2022) provides a much larger dataset, comprising 587 hours of speech from online teaching scenarios. While it is open-source and valuable for acoustic modeling, its domain-specific nature introduces biases in discourse structure, grammar, and lexical choices, making it less representative of everyday spontaneous conversations.

Spontaneous CS datasets, in contrast, are essential for modeling real-world language use but present greater challenges in collection and annotation. SEAME (Lyu et al., 2010) provides approximately 30 hours of spontaneous Mandarin-English conversations from 92 speakers in Singapore and Malaysia. It includes word-level transcriptions with time-aligned language boundaries, making it a valuable resource for code-switching research. Li et al. (2012) compiled 36 hours of spontaneous CS speech across various settings, including conversational meetings and student interviews, but only part-of-speech data is transcribed, limiting its usability for ASR research. ASCEND (Lovenia et al., 2022) provides a smaller (10.62 hours) yet fully transcribed and open-source dataset of spontaneous CS conversations recorded in Hong Kong, featuring 23 bilingual speakers.

Despite these advancements, most existing datasets exhibit limitations in scale, availability, or annotation completeness. Many either focus on isolated code-switching utterances rather than full dialogues, or remain inaccessible to the research community. In contrast, our dataset aims to bridge these gaps by providing 104 hours of spontaneous Mandarin-English CS speech, featuring full-length dialogue recordings with comprehensive transcriptions. It captures naturalistic code-switching patterns within extended conversations, making it a valuable resource for both ASR research and broader linguistic analysis.

While our focus is on Mandarin-English, it is important to acknowledge the growing body of research on code-switching in other language pairs. Notable examples include datasets and studies for Spanish-English (García et al., 2018), Arabic-English (Chowdhury et al., 2021), Hindi-English (Dey & Fung, 2014), and Manipuri-English (Singh et al., 2024), each contributing to a broader understanding of this complex linguistic phenomenon.

## 3 DATASET CREATION

The creation of the CS-Dialogue dataset involved a meticulous multi-stage process, encompassing careful data acquisition and rigorous annotation ensuring the development of a high-quality resource for code-switching research.

### 3.1 DATA ACQUISITION

#### 3.1.1 SPEAKER SELECTION

All speakers were native Chinese citizens with demonstrated fluency in English. Selection criteria prioritized individuals with significant exposure to English-speaking environments, such as overseas experience or high scores on standardized English proficiency tests (e.g., IELTS 6 or TEM-4). Prospective speakers underwent an audition to ensure adequate speech quality and language proficiency before being included in the recording sessions.

#### 3.1.2 ETHICAL CONSIDERATIONS AND COMPENSATION

Prior to participation, all speakers provided informed consent, granting permission for the collection, processing, and potential sharing of their data, including with parties located outside of China. The consent process adhered to ethical guidelines and ensured participants were fully aware of the data's intended use. Each speaker received financial compensation of 300 RMB (approximately 50 USD ) for their contribution to the dataset.

#### 3.1.3 TOPIC SELECTION

The dataset incorporates seven prevalent topics of daily relevance: personal topics, entertainment, technology, education, job, philosophy, and sports. A detailed overview of these topics could be found in Appendix A.3. To ensure comprehensive coverage, a minimum of 15 distinct speaker pairs engaged in discussions for each topic. Individual speaker pairs selected between two and six topics based on their personal interests, aiming to foster natural and engaging conversations.

#### 3.1.4 DIALOGUE RECORDING PROCEDURE

To facilitate natural and spontaneous interaction, paired dialogues were conducted through an audio-visual platform. Participants recorded their individual audio streams using smartphone microphones in quiet environments. For privacy and efficiency, only the audio recordings were retained for the dataset. A timekeeper facilitated each session, ensuring adherence to the established recording protocol. Each dialogue commenced with brief introductory remarks, transitioning into discussions centered on the pre-selected topics. The linguistic composition of the dialogue progressed systematically: initially in Mandarin Chinese, followed by a period of code-switching between Chinese and English, and concluding with exclusive use of English. Each topic segment was designed to last approximately 20 minutes, with a target allocation of 8 minutes for Chinese, 6 minutes for code-switching, and 6 minutes for English.

Table 2: Annotation Symbols and Definitions

| Symbol | Definition |
|---|---|
| `**` | Indicates unintelligible words or phrases. |
| `<FIL/>` | Filled pauses resulting from hesitation. |
| `<SPK/>` | Speaker-related noises, such as lip smacking, laughter, coughing, or throat clearing. |
| `<NON/>` | Non-speech noises, such as door slams, knocks, or ringing sounds. |
| `<NPS/>` | Noises made by individuals other than the designated speakers, including speech or noise. |

While the timekeeper provided prompts to maintain the intended schedule, natural variations in pacing were permitted to encourage spontaneous and authentic communication. Participants were not strictly limited to a single language during any segment. They could code-switch naturally in monolingual phases, and monolingual speech was also allowed during the code-switching segment. This flexible setup helped preserve spontaneity. The transcriptions faithfully reflect what was actually spoken, including deviations from the intended language schedule, ensuring the dataset captures authentic conversational behavior. A dedicated observer monitored each session, verifying procedural compliance and recording relevant metadata for subsequent analysis. The entire procedure took approximately 1.5 hours. All audio files in the dataset are stored in a 16 kHz, 16-bit, mono, PCM WAV format.

## 3.2 ANNOTATION

To ensure high data quality and support downstream tasks, all audio files underwent a rigorous annotation process. This included precise manual transcription, detailed labeling of non-lexical events, and strict quality control procedures. All annotations were carried out by a dedicated in-house team (see Appendix A.1 for annotator details) following a standardized protocol. An illustrative example of a dialogue transcription is provided in Appendix A.2.

The transcription process prioritized accurate representation of the spoken content, focusing on the speaker's actual pronunciation. The following guidelines were implemented to maintain consistency and ensure high transcription quality:

1. **Word Count Fidelity:** Transcriptions were required to maintain a precise word-for-word correspondence with the spoken utterance, preventing both omissions and additions.

2. **Treatment of Disfluencies:** Clear repetitions of sounds or words were transcribed verbatim (e.g., "放放假" transcribed as "放放假"). Partially articulated syllables were transcribed using the most appropriate homophone (e.g., "放假" pronounced as "fu-fang4-jia4" transcribed as "夫放假"). Epenthetic or extremely faint sounds were disregarded.

3. **Numerical Representation:** Arabic numerals were converted to their corresponding Chinese characters or English words, depending on the context and pronunciation (e.g., "711" transcribed as "七幺幺" or "Seven Eleven").

4. **Accent Accommodation:** Regional accents and variations in pronunciation (e.g., distinctions between retroflex and non-retroflex consonants, nasal finals, or the pronunciation of /h/ and /f/ or /l/ and /n/) were preserved in the transcription without correction.

5. **Punctuation Conventions:** Punctuation marks, including both Chinese and English symbols, were applied according to standard grammatical conventions and semantic context to ensure clarity and accurate segmentation.

6. **Spelling conventions:** Spelling followed common English conventions and standards to ensure quality of annotations.

7. **Acronym Representation:** Acronyms were transcribed using uppercase letters separated by spaces (e.g., "I B M"). Utterances consisting of three or fewer letters transcribed as an acronym were categorized as Chinese.

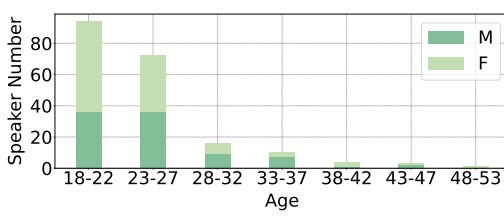 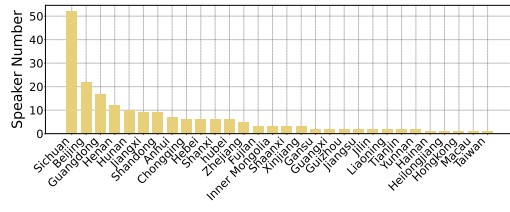

(a) Distribution of speakers by age and gender    (b) Distribution of speakers by region

Figure 1: Demographic distributions of the speaker population

In addition to the transcription of spoken words, a set of specialized symbols was used to annotate non-lexical events and acoustic phenomena. These symbols, detailed in Table 2, provided additional information about the acoustic characteristics of the data.

Following the initial annotation, a separate quality control team performed a rigorous review process to ensure data accuracy. Discrepancies were resolved through discussion and iterative refinement of the annotation protocol, ensuring the high transcription quality.

# 4    DATASET DESCRIPTION

This section provides a comprehensive overview of the CS-Dialogue dataset, including its profile, statistical analysis, and details on speaker demographics, duration, topic distribution, and textual characteristics.

## 4.1    PROFILE

The CS-Dialogue dataset comprises 104.02 hours of spontaneous Mandarin-English code-switching speech from 200 speakers, structured as 100 dialogues (200 raw recordings, as each dialogue involves two participants). These dialogues encompass 320 topic sessions, offering a diverse range of conversational contexts. The dataset contains 38,917 utterances. Table 3 summarizes the key characteristics of the dataset, including the total duration, number of speakers, dialogues, utterances, and language distribution.

For model development and evaluation, the dataset is divided into three speaker-independent sets: training, development, and test. The breakdown of each split is presented in Table 4. Critically, these splits are speaker-independent ensuring a robust evaluation of model generalization.

Table 3: Overview of our dataset

| Characteristic | Value |
|---|---|
| Duration (hrs) | 104.02 |
| # Speakers | 200 |
| # Dialogues | 100 |
| # Raw Recordings | 200 |
| # Topic Sessions | 320 |
| # Utterances | 38,917 |
| Avg. Duration (s) | 9.62 |

## 4.2    STATISTICS

### 4.2.1    SPEAKER DEMOGRAPHICS

The age and gender distribution of the speakers is illustrated in Figure 1a. Speaker ages range from 18 to 53, grouped into four-year intervals. Male speakers are represented in green and female speakers in light green in the stacked bar chart. A notable trend is the concentration of speakers in the younger age brackets (18-22 and 23-27), with a relatively balanced gender distribution. The decrease in speaker numbers in older age groups may be attributed to the greater prevalence of Mandarin-English bilingualism among younger generations, or potential challenges in recruiting older participants with the required language proficiency. The data was collected from various regions in China.

Table 4: Summary of data splits

| Split | # Spk. | # Utt. | Dur. (hrs) | Avg. (s) |
|---|---|---|---|---|
| Train | 140 | 26,428 | 68.97 | 9.40 |
| Dev | 30 | 6,196 | 18.30 | 10.63 |
| Test | 30 | 6,293 | 16.74 | 9.58 |
| Total | 200 | 38,917 | 104.02 | 9.62 |

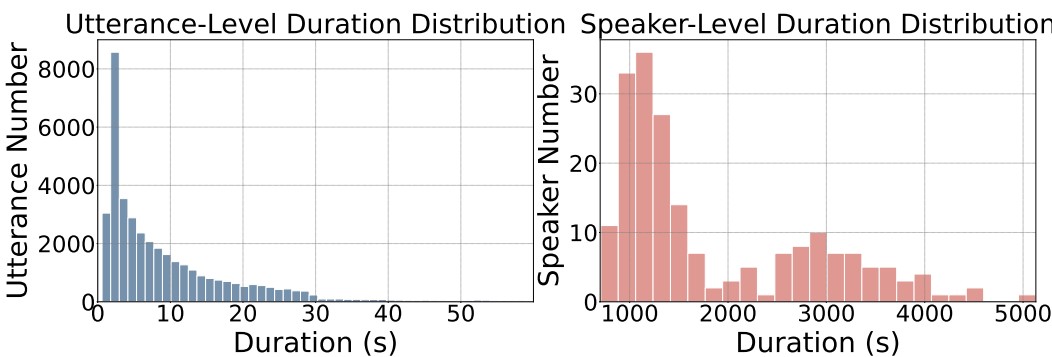

Figure 2: Utterance-level (left) and speaker-level (right) duration distributions.

The regional distribution of speakers, based on their reported origin, is displayed in Figure 1b. Sichuan has the highest representation, followed by Beijing and Guangdong, while the remaining regions have significantly fewer speakers.

### 4.2.2 DURATION ANALYSIS

Utterance-level and speaker-level duration distributions are presented in Figure 2. Most utterances are under 30 seconds, and the majority of speakers have a total speaking time clustered towards the lower end of the range. However, a few speakers contribute significantly more data, leading to a long-tailed distribution.

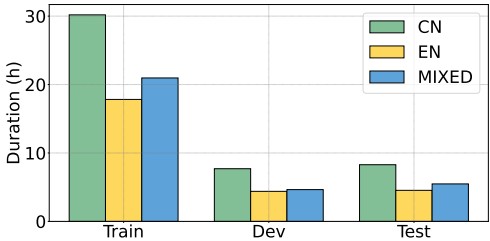

Figure 3: Duration of each language per data split

The training, development, and test sets exhibit a consistent proportional distribution of Chinese, English, and mixed-language durations, as shown in Figure 3. This balanced representation of each language category within each split ensures that models trained on one split are likely to generalize well to others. Appendix B.1 details the distribution of full-dialogue durations.

### 4.2.3 DIALOGUE TOPIC ANALYSIS

The dataset's conversation topics are distributed as shown in Table 5. Categorized into seven broad themes—Personal Topics, Entertainment, Technology, Education, Job, Philosophy, and Sports—the 320 topic sessions offer diverse conversational contexts (see Appendix A.3 for details). Personal Topics are the most frequent (24.38%; 78/320 sessions), while Philosophy is the least frequent (4.69%), indicating a focus on everyday conversational themes, along with a smaller, yet still significant, representation of more specialized topics.

Further details on the distribution of these seven conversation topics across each data split (training, development, and test sets) are provided in Appendix B.2.

Table 5: Topic distribution in the full-dialogue recordings

| Topic Name | Frequency | Proportion |
|---|---|---|
| Personal topics | 78 | 24.38% |
| Entertainment | 60 | 18.75% |
| Technology | 26 | 8.13% |
| Education | 61 | 19.06% |
| Job | 36 | 11.25% |
| Philosophy | 15 | 4.69% |
| Sports | 44 | 13.75% |
| Total | 320 | 100.00% |

### 4.2.4 TEXT ANALYSIS

An analysis of frequent strings (Appendix C) reveals distinct patterns in language use and code-switching strategies. Discourse markers like "我觉得" (I think) and "比如说" (for example) characterize the Chinese segments, while phrases like "a lot of" and "I think it's" are frequently used in the English segments. A crucial observation in the mixed-language segments is the frequent use of function words from one language to frame content words from the other (e.g., "you know 就"), suggesting that code-switching commonly occurs at clause or phrase boundaries.

## 5 EXPERIMENTS

This section presents our experimental evaluation of the CS-Dialogue dataset. We assess the performance of various ASR models, including those trained from scratch and pre-trained models, with and without fine-tuning on our data.

### 5.1 METRICS

ASR performance on the code-switching dataset is evaluated using three metrics: Mixture Error Rate (MER), Word Error Rate (WER), and Character Error Rate (CER). Following (Shi et al., 2020), MER is adopted as the primary metric due to its holistic assessment of ASR accuracy, calculating the edit distance considering both Chinese characters and English words. In addition to MER, WER and CER are calculated separately for English and Chinese segments to provide more granular insights into per-language performance.

### 5.2 BASELINE MODELS

Two categories of baseline ASR models are evaluated: models trained from scratch on the CS-Dialogue dataset and models pre-trained on large external datasets. Details regarding model training and hyperparameter configurations are provided in Appendix D.

### 5.2.1 MODELS TRAINED FROM SCRATCH

We train three ASR models from scratch using the WeNet toolkit (Yao et al., 2021): (1) **Transformer** (Vaswani, 2017), an attention-based encoder-decoder (AED) model; (2) **Conformer** (Gulati et al., 2020), which integrates convolution and self-attention for modeling local and global context; and (3) **Branchformer** (Peng et al., 2022), which introduces a branching mechanism to capture diverse speech patterns. All models are trained solely on the CS-Dialogue training set using a joint CTC (Graves et al., 2006) and AED (Chorowski et al., 2014) loss, without external data.

### 5.2.2 PRE-TRAINED MODELS

Several state-of-the-art pre-trained models are also evaluated on the CS-Dialogue dataset:

- **Whisper** (Radford et al., 2023): A robust, multilingual Transformer-based ASR model pre-trained by OpenAI on 680,000 hours of diverse speech data[1].
- **Qwen2-Audio** (Chu et al., 2024): A large-scale audio-language model from Alibaba[2], capable of processing various audio inputs and performing tasks like audio analysis and speech-instruction following.
- **SenseVoice-Small** (An et al., 2024): A non-autoregressive, encoder-only speech foundation model from Alibaba designed for multilingual, multi-style ASR and other speech understanding tasks[3].
- **FunASR-Paraformer** (Gao et al., 2022): A fast and accurate non-autoregressive (NAR) end-to-end ASR model[4].

---

[1]`https://github.com/openai/whisper`
[2]`https://github.com/QwenLM/Qwen2-Audio`
[3]`https://github.com/FunAudioLLM/SenseVoice`
[4]`https://github.com/modelscope/FunASR`

Table 6: Performance of different models training from scratch under various decoding strategies.

| Model | # Params | Greedy | | | Beam | | | Attention | | | Attention Rescoring | | |
|---|---|---|---|---|---|---|---|---|---|---|---|---|---|
| | | CER | WER | MER | CER | WER | MER | CER | WER | MER | CER | WER | MER |
| Transformer | 29M | 22.56 | 45.34 | 27.21 | 22.24 | 45.19 | 27.01 | 39.23 | 62.80 | 44.05 | 21.60 | 43.44 | 26.06 |
| Branchformer | 29M | 18.86 | 39.16 | 23.01 | 18.78 | 39.20 | 22.95 | 44.06 | 60.90 | 47.50 | 18.29 | 37.55 | 22.23 |
| Conformer | 31M | **15.91** | **33.67** | **19.54** | **15.88** | **33.60** | **19.50** | 24.98 | 42.75 | 28.61 | **15.45** | **32.36** | **18.91** |

Table 7: Performance comparison of different ASR models on the CS-Dialogue test set. S: Substitution; D: Deletion; I: Insertion.

| Model | # Param | CER (%) | WER (%) | MER (%) | | | |
|---|---|---|---|---|---|---|---|
| | | | | S | D | I | Overall |
| Whisper Large-V2 | 1,550M | 10.70 | 31.11 | 6.00 | 7.60 | 1.69 | 15.29 |
| Qwen2-Audio | 8.2B | 7.15 | 19.82 | 4.32 | 1.82 | 3.62 | 9.76 |
| Paraformer | 220M | **3.70** | 32.02 | 6.30 | **0.98** | 2.37 | 9.65 |
| SenseVoice-Small | 234M | 4.42 | **15.57** | **3.44** | 1.42 | **1.85** | **6.71** |

Table 8: Zero-shot and fine-tuning performance of different Whisper models and SenseVoice-Small on the CS-Dialogue test set.

| Model | # Param | Zero-shot | | | Fine-tuning | | |
|---|---|---|---|---|---|---|---|
| | | CER (%) | WER (%) | MER (%) | CER (%) | WER (%) | MER (%) |
| Whisper-Tiny | 38M | 27.83 | 41.69 | 31.11 | 19.24 | 29.64 | 21.38 |
| Whisper-Base | 74M | 19.90 | 37.21 | 23.90 | 15.36 | 27.20 | 17.80 |
| Whisper-Small | 244M | 12.82 | 30.81 | 16.76 | 7.51 | 16.09 | 9.26 |
| Whisper-Medium | 769M | 11.34 | 32.57 | 15.88 | 6.12 | 13.02 | 7.53 |
| SenseVoice-Small | 234M | **4.42** | **15.57** | **6.71** | **3.34** | **10.87** | **4.99** |

## 5.3 RESULT ANALYSIS

### 5.3.1 PERFORMANCE OF MODELS TRAINED FROM SCRATCH

The performance comparison of models trained from scratch is presented in Table 6. Across all decoding methods (greedy decoding, beam search, attention decoding, and attention rescoring), the Conformer consistently outperforms both the Transformer and Branchformer. Attention rescoring yields the best performance for all models, resulting in the lowest CER, WER, and MER. For instance, the Conformer achieves a CER of 15.45%, a WER of 32.36%, and an MER of 18.91% with attention rescoring, a substantial improvement over the results obtained with greedy decoding (15.91% CER, 33.67% WER, 19.54% MER). While the Branchformer generally surpasses the Transformer in performance, it exhibits the highest error rate under the attention decoding strategy.

### 5.3.2 PERFORMANCE OF PRE-TRAINED MODELS

Table 7 presents the performance of several pre-trained models on the test set. Among them, SenseVoice-Small achieves the lowest MER (6.71%). Despite its broader capabilities and significantly larger size, Qwen2-Audio reports a higher MER (9.76%) compared to SenseVoice-Small. Similarly, Whisper Large-V2, another large-scale multilingual model, exhibits the highest error rates, with an MER of 15.29%. SenseVoice-Small achieves better MER despite its small size likely because it is optimized for ASR in a limited set of languages, with a task-specific architecture and substantial exposure to Chinese during training. In contrast, larger models like Qwen2-Audio and Whisper are trained for a wide range of tasks and languages, which may dilute their performance on specialized CS-ASR scenarios. Notably, all models show a higher proportion of substitution errors relative to deletions or insertions, as revealed by the MER breakdown.

Among the pre-trained models, Whisper is one of the most widely adopted ASR foundation models. We evaluate different sizes of Whisper in both zero-shot and fine-tuned settings, and additionally include SenseVoice-Small, which achieves the best zero-shot performance. The results are presented in Table 8. Fine-tuning consistently yields substantial improvements across all Whisper model sizes. Within the Whisper family, the Medium model achieves the best post-finetuning performance. However, the overall best results are obtained by SenseVoice-Small after fine-tuning, reaching a CER of 3.34%, a WER of 10.87%, and an MER of 4.99%. These findings demonstrate that while larger Whisper models benefit more from fine-tuning, SenseVoice-Small sets the performance benchmark for code-switching ASR in our experiments.

Beyond the quantitative results, a qualitative analysis of the Whisper-Medium model's output is provided in Appendix E. This analysis includes example transcriptions, comparing zero-shot and fine-tuned performance, and highlights common error types.

### 5.3.3 IMPACT OF DIALOGUE CONTEXT

To investigate the benefit of utilizing full dialogue context, a characteristic feature of the CS-Dialogue dataset, we conducted an additional experiment with the Whisper Large-V2 model. Specifically, we evaluated its performance on CS-Dialogue while varying the number of preceding dialogue turns provided as contextual prompts. The results, presented in Table 4, demonstrate a clear trend: increasing the amount of dialogue context significantly improves code-switching ASR performance.

Figure 4: Impact of dialogue context on Whisper Large-V2 performance

| Context Segments | CER (%) | WER (%) | MER (%) |
|---|---|---|---|
| 0 (baseline) | 10.70 | 31.11 | 15.29 |
| 1 | 10.30 | 29.34 | 14.51 |
| 2 | 9.81 | 28.05 | 13.74 |
| 3 | **9.13** | **26.26** | **12.97** |

For instance, using three preceding dialogue turns as context reduces the MER from 15.29% (no context) to 12.97%. This finding highlights the advantage of CS-Dialogue's full dialogue structure over datasets comprising only isolated utterances.

### 5.3.4 TOPIC-SPECIFIC PERFORMANCE ANALYSIS

Figure 5 illustrates the MER of the four pre-trained ASR models across the seven conversation topics. Model performance varies considerably across topics. SenseVoice-Small consistently achieves the lowest MERs, indicating its superior performance on this task. Comparing Qwen2-Audio and Paraformer reveals no consistent dominance of one model over the other; instead, their relative performance is topic-dependent. In addition, "Sports" and "Philosophy" tend to have higher MERs for all models, while "Job" and "Technology" generally exhibit lower MERs, suggesting varying levels of difficulty across topics.

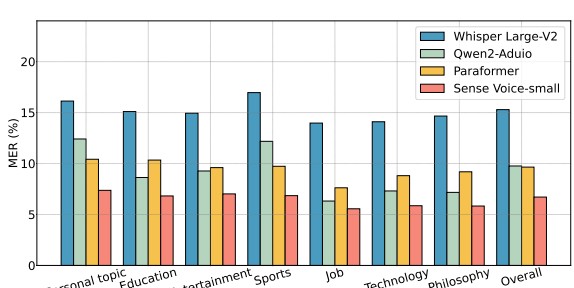

Figure 5: Comparison of MER for Whisper Large-V2, Qwen2-Audio, Paraformer, and SenseVoice-Small across different conversation topics

## 6 CONCLUSION

In this paper, we presented CS-Dialogue, a new 104-hour large-scale dataset of spontaneous Mandarin-English code-switching dialogues. Unlike most existing datasets that primarily offer isolated utterances, CS-Dialogue provides full-length dialogue recordings and complete transcriptions, enabling unprecedented research into the contextual dynamics of code-switching. This resource addresses existing dataset limitations by capturing naturalistic code-switching patterns. Our rigorous data creation and baseline experiments highlight code-switching challenges and the value of fine-tuning. CS-Dialogue offers a benchmark for future ASR and dialogue modeling, aiming to advance robust multilingual communication systems that can leverage conversational context. Future extensions could explore additional language pairs and more diverse conversational settings.

ETHICS STATEMENT

The collection and use of the CS-Dialogue dataset were conducted in accordance with established ethical guidelines and regulations for human subjects research. Prior to participation, all speakers were provided with a comprehensive information sheet detailing the study's purpose, data collection procedures, and their rights as participants, including the right to withdraw from the study at any time without penalty. Informed consent was obtained from each speaker, explicitly authorizing the recording of their conversations, the processing and analysis of their speech data, and the potential sharing of anonymized data with other researchers (including those located outside of China) for research purposes.

Participants were assured that their data would be treated with strict confidentiality and anonymized to protect their privacy. No personally identifiable information (e.g., names, specific locations) will be included in the released dataset or any associated publications. Participants received compensation for their time and contribution to the study, commensurate with standard rates for similar research participation. The research protocol, including the informed consent process and compensation procedures, was designed to ensure the protection of participants' rights and well-being. To mitigate potential risks, the topics of discussion during the dialogues were carefully selected to avoid sensitive or potentially harmful content. Participants were given the autonomy to choose topics from a predefined list and were free to pause or stop the recording at any point during the session.

REPRODUCIBILITY STATEMENT

To promote reproducibility and facilitate future research, we will publicly release the CS-Dialogue dataset under a permissive license for non-commercial use. The dataset includes detailed transcriptions, annotations, and metadata, enabling researchers to fully replicate our experiments and explore new directions. In addition, we have reported the training configurations and hyperparameters of our baseline models, which are implemented using open-source toolkits.

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

# A DATASET DETAILS

## A.1 ANNOTATOR INFORMATION

Table A.1 provides a breakdown of the annotators' demographic characteristics, including their age, gender, hometown, and educational background.

## A.2 DIALOGUE TRANSCRIPTION FORMAT

Dialogue transcription format are shown in Figure A.2 as an example. Note that he names used in this example (e.g., "凯丽", "贝拉") are pseudonyms and do not correspond to the real names of the speakers, ensuring the privacy of participants.

```
xmin = 0
xmax = 6148.265
tiers? <exists>
size = 3
item []:
 item [1]:
    class = "IntervalTier"
    name = "ZH-CN_U0018_S0"
    xmin = 0
    xmax = 6148.265
    intervals: size = 368
    intervals [1]:
      xmin = 0
      xmax = 5.81
      text = ""
    intervals [2]:
      xmin = 5.81
      xmax = 11.232
      text = "嗨，嗯，你好啊，你好漂亮啊，嗯。"
    intervals [3]:
      xmin = 11.232
      xmax = 12.6
      text = ""
    intervals [4]:
      xmin = 12.6
      xmax = 20.415
      text = "嗯，哦，你可以叫我凯丽，凯丽就行，
嗯，那你叫什么名字呢？"
    intervals [5]:
      xmin = 20.415
      xmax = 21.155
      text = ""
    intervals [6]:
      xmin = 21.155
      xmax = 25.245
      text = "啊，好，嗨，贝拉，嗯。"
```

Figure A.1: A format example of dialogue transcription file (TextGrid)

## A.3 TOPICS, DESCRIPTIONS, AND EXAMPLES

Table A.2 provides details on the seven conversation topics covered in the dataset, including a brief description of each topic and an example utterance illustrating typical content and code-switching patterns. This information clarifies the thematic scope of the data and provides context for interpreting the experimental results.

Table A.1: Summary of Annotator Demographics

| Category | Value | Count/Percentage |
|---|---|---|
| Gender | Male | 6 (40%) |
| | Female | 9 (60%) |
| Education | Bachelor's | 12 (80%) |
| | Master's | 3 (20%) |
| Hometown | Liaoning | 1 (6.67%) |
| | Henan | 1 (6.67%) |
| | Shanxi | 3 (20%) |
| | Zhejiang | 1 (6.67%) |
| | Jiangxi | 2 (13.33%) |
| | Beijing | 1 (6.67%) |
| | Fujian | 1 (6.67%) |
| | Hebei | 3 (20%) |
| | Shaanxi | 1 (6.67%) |
| | Ningxia | 1(6.67%) |
| Age | 21 | 5 (33.33%) |
| | 22 | 1 (6.67%) |
| | 24 | 4 (26.67%) |
| | 25 | 3 (20%) |
| | 28 | 2 (13.33%) |

Table A.2: Details of topics, descriptions, and examples

| Topic | Description | Example |
|---|---|---|
| Personal | Discussions centered on individual experiences, preferences, and relationships. | "就是我听你的描述，感觉你喜欢 Taylor. 因为我其实我有个弟弟也很喜欢 Taylor，就是但是他性格还确实跟你相差蛮大，就是你给我的感觉 You are very a quiet boy" |
| Entertainment | Conversations focusing on various forms of entertainment and cultural trends. | "对，因为我们都想表现的自己非常的 courage, 但其实我小的时候也看着也非常 frighten，然后我会直接¡FIL¿放到 home 之后就一直坐坐在 sofa 上面看到十点都 can't move" |
| Technology | Debates and dialogues concerning technological advancements and their impact. | "是的，而且他们会通过算法去非常精准地知道你到底想要看一些什么样的东西，所以我感觉其实有的时候 big data 也是一个非常恐怖的东西" |
| Education | Discussions about the academic environment, including challenges and experiences. | "那是你们这个 group 自己去想一个 topic 呢，还是这个 professor 会提供一些他的 project 来支撑你们的毕业论文" |
| Job | Conversations about past or present employment situations, work environment, co-workers, etc. | "对对，是如果有有更多的 opportunity 去供我去选择的话，我还是可能会就是放开专业去选择更多的这个看一看，开阔一下我的 horizon" |
| Sports | Discussions on sports activities, athletes, benefits of exercise, etc. | "但是这种持续的时间 I couldn't find very long，就是我也能找到那种 feeling， but very quickly， It disappear only maybe half an hour is a longest period， sometimes 也就十五分钟二十分钟" |
| Philosophy | Discussions about philosophical ideas and debates on current social issues. | "这就是他们现在所处的一个 dilema, It's really classic， I feel like it's happening everywhere。因为每一个国家都有他各自的 minorities，然后也不得不承认有些地方的 educational resource 真的没有另一些地方更加的 advanced，更加的丰富。" |

## B    Dialogue analysis

### B.1    Full-dialogue duration distribution

As presented in Figure B.1, most of full-dialogue recordings are between 2000 to 3000 seconds. This distribution indicates a dataset primarily composed of relatively shorter full-dialogue recordings, with a smaller number of significantly longer recordings.

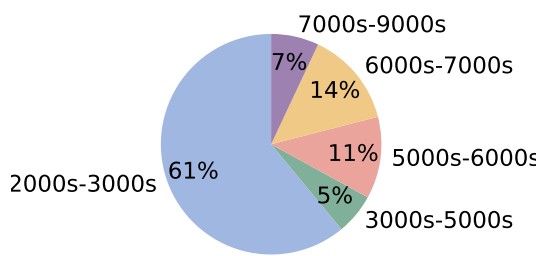

Figure B.1: Duration distribution of full-dialogue recordings

### B.2    Topic distribution

The distribution of the seven conversation topics across each data split is detailed in Table B.1. This table presents, for each topic, the total duration, its proportion of the entire dataset, the utterance count, total duration, and average utterance length for each split. While the topic distribution is relatively consistent across the three sets, some variation exists in average utterance lengths. Notably, "Philosophy" tends to have slightly longer utterances than other categories, particularly in the test set (12.95s).

Table B.1: Topic distribution across training, development, and test sets: counts, durations, and average utterance lengths

| Topic | Dur. (hrs) | Proportion | Train | | | Dev | | | Test | | |
|---|---|---|---|---|---|---|---|---|---|---|---|
| | | | Count | Dur. (hrs) | Avg. (s) | Count | Dur. (hrs) | Avg. (s) | Count | Dur. (hrs) | Avg. (s) |
| Personal topics | 21.53 | 20.69% | 5,865 | 14.59 | 8.95 | 1,348 | 3.55 | 9.48 | 1,487 | 3.39 | 8.22 |
| Education | 19.20 | 18.45% | 4,853 | 13.34 | 9.9 | 1,014 | 3.11 | 11.04 | 931 | 2.75 | 10.63 |
| Entertainment | 23.88 | 22.95% | 6,609 | 15.85 | 8.63 | 1,509 | 4.39 | 10.47 | 1,402 | 3.64 | 9.34 |
| Sports | 14.14 | 13.59% | 3,476 | 8.64 | 8.95 | 745 | 2.35 | 11.34 | 1,236 | 3.15 | 9.17 |
| Job | 11.94 | 11.48% | 2,555 | 7.37 | 10.38 | 1,052 | 3.02 | 10.35 | 513 | 1.55 | 10.9 |
| Technology | 8.70 | 8.36% | 2,151 | 6.09 | 10.19 | 335 | 1.24 | 13.36 | 475 | 1.37 | 10.35 |
| Philosophy | 4.65 | 4.47% | 919 | 3.11 | 12.17 | 193 | 0.64 | 11.92 | 249 | 0.9 | 12.95 |

## C    Text analysis

To illustrate common linguistic patterns and code-switching behaviors, Table C.1 presents the most frequent strings found in the Chinese (CN), English (EN), and mixed-language utterances within the dataset. The table lists the strings and their corresponding frequencies.

## D    Experimental configurations

This section provides detailed hyperparameters used for training and fine-tuning ASR models discussed in the paper. All experiments were conducted using four GTX 3090 for several hours. All models utilized in this research are open-source and operate under the MIT License.

Table C.1: Top Frequent Strings in Each Language Category

| CN | | EN | | MIXED | |
|---|---|---|---|---|---|
| String | Count | String | Count | String | Count |
| 我觉得 | 3,391 | a lot of | 304 | you know 就 | 20 |
| 的时候 | 2,056 | I think it's | 208 | know 就是 | 18 |
| 比如说 | 1,410 | I want to | 182 | 这个 AI | 14 |
| 是一个 | 1,173 | do you like | 178 | school 的时 | 12 |
| 因为我 | 1,119 | do you have | 168 | 我的 friends | 11 |
| 然后我 | 1,033 | I don't know | 143 | 就是 you | 11 |
| 的一个 | 982 | yeah yeah yeah | 142 | 就是 AI | 10 |
| 但是我 | 958 | and I think | 141 | 一个 very | 10 |
| 的一些 | 925 | so I think | 139 | 非常 interesting | 10 |
| 就是我 | 913 | yeah I think | 132 | I think 我 | 9 |
| 的就是 | 809 | what do you | 118 | 常的 happy | 9 |
| 非常的 | 792 | go to the | 115 | play 麻将 | 9 |
| 或者是 | 733 | but I think | 110 | high school 的 | 8 |
| 有一些 | 730 | I think I | 108 | 一个 big | 8 |
| 我感觉 | 726 | you want to | 107 | together 然后 | 7 |

## D.1 TRAINING ASR MODEL FROM SCRATCH

Table D.1 presents the training hyperparameters for Transformer, Branchformer and Conformer using Wenet toolkit, including batch size, learning rate and epochs. A dynamic batch size is utilized, constrained to a maximum of 60,000 frames per batch.

Table D.1: Hyperparameters for training ASR models from scratch.

| Model | Batch size | Learning rate | Epochs |
|---|---|---|---|
| Transformer | Dynamic | 1.00E-03 | 150 |
| Branchformer | Dynamic | 1.00E-03 | 150 |
| Conformer | Dynamic | 1.00E-03 | 150 |

## D.2 FINE-TUNING ASR MODEL

Table D.2 presents the hyperparameters used during fine-tuning of the different Whisper model versions and SenseVoice-Small. These parameters include the learning rate and the number of epochs. A dynamic batch size is utilized, constrained to a maximum of 12,000 frames per batch.

Table D.2: Hyperparameters for fine-tuning different Whisper versions and SenseVoice-Small.

| Model | Batch size | Learning rate | Epochs |
|---|---|---|---|
| Whisper-Tiny | 16 | 1.00E-05 | 20 |
| Whisper-Base | 16 | 1.00E-05 | 20 |
| Whisper-Small | 16 | 1.00E-05 | 20 |
| Whisper-Medium | 16 | 1.00E-05 | 20 |
| SenseVoice-Small | Dynamic | 4.00E-05 | 10 |

## E CASE STUDIES

To illustrate the types of errors made by the Whisper Medium model and the improvements achieved through fine-tuning, Figure E.1 presents example transcriptions for several utterances. The figure compares the zero-shot and fine-tuned outputs against the ground truth transcriptions, highlighting differences and providing the associated MER. We obsever that whisper designed for both ASR and S2TT tasks, exhibits an unintended behavior in code-switching ASR scenarios. Specifically, the model occasionally produces translations of the input speech rather than accurate transcriptions, deviating from the expected ASR output.

```
Utterance: ZH-CN_U1093_S0_65.wav
Ground truth:    然 后 就 还 直 直 接 就 是 FALL IN     THE STREET
Zero-shot:       然 后 就 还 直 - 接 就 要 FOR INDUSTRY 去     MER: 46.15 % N=13 C=7 S=4 D=2 I=0
Fine-tuning:     然 后 就 还 直 - 接 就 是 FOUR IN THE STREET MER: 15.38 % N=13 C=11 S=1 D=1 I=0

Utterance: ZH-CN_U0017_S0_2.wav
Ground truth:    HELLO HELLO 很 高 兴 认 识 你 啊
Zero-shot:       哈 喽 哈 喽 很 高 兴 认 识 你 啊           MER: 44.44 % N=9 C=7 S=2 D=0 I=2
Fine-tuning:     HELLO HELLO 很 高 兴 认 识 你 呀           MER: 11.11 % N=9 C=8 S=1 D=0 I=0

Utterance: ZH-CN_U1093_S0_175.wav
Ground truth:    国 内 的 SINGER 的 话 I MOST LIKE IS 周 杰 伦 DO YOU KNOW
Zero-shot:       国 内 的 SINGER 的 话 I MOST LIKE IS 周 杰 伦 对 吗         MER: 18.75 % N=16 C=13 S=2 D=1 I=0
Fine-tuning:     国 内 的 SINGER 的 话 I MOST LIKE IS 周 杰 伦 DO YOU KNOW  MER: 0.00 % N=16 C=16 S=0 D=0 I=0

Utterance: ZH-CN_U0066_S0_60.wav
Ground truth:    哦 我 平 常 喜 欢 做 运 动 SOMETHING LIKE BASKETBALL FOOTBALL TABLE TENNIS AND SWIMMING 然 后   我 也 经 常 去 健 身 房
                 和 一 些 STRONG MAN     交 流 一 下 运 动 技 巧 然 后 我 最 喜 欢 的 运 动 应 该 是 BASKETBALL
Zero-shot:       - 我 平 常 喜 欢 做 运 动 比     如 球     足     球 乒 乓 球     和 游 泳 我 也 经 常 去 健 身 房 和 一 些 力 量 男 生 交 流 运
                 动 技 巧 我 最 喜 欢 的 运 动 应 该 是 球
                                                         MER: 39.62 % N=53 C=35 S=13 D=5 I=3
Fine-tuning:     - 我 平 常 喜 欢 做 运 动 SOMETHING LIKE BASKETBALL FOOTBALL TABLE TENNIS AND SWIMMING 然 后 我 也 经 常 去 健 身 房
                 和 一 些 STRONG MAN 交 流 一 下 运 动 技 巧 然 后 我 最 喜 欢 的 运 动 应 该 是 BASKETBALL
                                                         MER: 1.89 % N=53 C=52 S=0 D=1 I=0
```

Figure E.1: Examples of ASR output from the Whisper Medium model under zero-shot and fine-tuned conditions, showing ground truth transcriptions and error rates

## F LIMITATIONS

While CS-Dialogue represents a significant contribution to the field, it has certain limitations. First, the dataset focuses exclusively on Mandarin-English code-switching. While this is a prevalent language pair, future work should expand to include other language combinations to enhance the generalizability of code-switching ASR models. Second, all participants are native Chinese speakers with strong English proficiency. The dataset does not include native English speakers who code-switch into Mandarin, which represents another important aspect of bilingual conversation. Third, although the dialogues are spontaneous, they are still recorded in a controlled environment, which may not fully reflect the acoustic diversity of real-world scenarios (e.g., noisy public spaces, varying microphone quality). Future work could explore data augmentation techniques to simulate a wider range of acoustic conditions.

## G LLMs USAGE

In this work, Large Language Models (LLMs) were used to assist with language refinement and manuscript polishing. Specifically, LLMs helped improve clarity, coherence, and grammar. We independently developed all research ideas, experiments, and conclusions. We take full responsibility for the content, ensuring it meets academic standards and avoids any form of misconduct or plagiarism.

