# OpenReview forum: "CS-Dialogue: A 104-Hour Dataset of Spontaneous Mandarin-English Code-Switching Dialogues for Speech Recognition"
_ICLR.cc/2026/Conference — Submitted to ICLR 2026_

### Official Review · Reviewer_M8ep · 2025-10-26

**Soundness:** 3
**Presentation:** 3
**Contribution:** 3
**Rating:** 4
**Confidence:** 4

**Summary:**

The paper introduces a large, high-quality dataset of real conversational Mandarin-English code-switching, addressing the lack of authentic resources in this area. It provides detailed speaker and topic diversity, and demonstrates that training ASR models with this data significantly improves code-switching recognition.

**Strengths:**

1. This work presents a large-scale, spontaneous Mandarin-English code-switching speech dataset, filling a notable gap in the field. The paper invested substantial effort in collecting and processing real conversational data, which is of great value to the research community. The dataset addresses the scarcity of authentic code-switching conversational data and will facilitate further advancements in ZH-EN CS ASR.
2. The paper provides detailed analyses of the dataset’s composition, including speaker demographics (gender, age, region), topic distribution, and utterance statistics. Such thorough documentation enhances the dataset’s utility and transparency for future research.
3. Experiments on SenseVoice-small and other state-of-the-art models show that fine-tuning with this dataset can effectively improve code-switching ASR performance. The results validate the practical value of the resource.

**Weaknesses:**

1. **Insufficient Detail on Code-Switching Text Construction** The one of most critical aspect of a code-switching dataset is how the code-switched utterances are generated. The paper does not provide sufficient detail on whether the code-switching segments were produced spontaneously by speakers or guided by prompts. If speakers were given too much freedom, there is a risk that some utterances may be unnatural or overly contrived. For example, in Table A.2, some “sports” examples contain long English segments that are rarely observed in natural code-switching. The authors should clarify the construction process and discuss its impact on data authenticity.
2. **Limited Duration of Mixed (Code-Switching) Data** According to Figure 3, the mixed-language (code-switching) portion comprises only about 30 hours, less than one-third of the total dataset.
3. **Lack of Comparison with Synthetic or Segmented Data Approaches** Previous studies have shown that training with high-quality TTS-synthesized code-switching data or segmented real utterances can substantially improve code-switching ASR performance. However, this paper does not include experiments comparing the effectiveness of real conversational data versus synthetic or segmented data. Such comparisons would help highlight the unique advantages of the presented dataset.

**Questions:**

This work represents a major effort and a valuable resource for the community. I sincerely appreciate the authors’ dedication and the significant investment of time and funding. However, I feel that it falls short of the standards expected for a technical paper at a research conference. Beyond the resource construction and baseline experiments, the paper lacks deeper research insights and thoughtful analysis that would help the community better understand the challenges and opportunities in code-switching speech recognition.

1. While the paper demonstrates the effectiveness of fine-tuning on SenseVoice-small, the test and training sets are from the same source. It would be valuable to know whether fine-tuning on CS-Dialogue leads to performance degradation on other open ASR benchmarks, indicating potential overfitting or lack of generalization.

2. Given the proven benefits of high-quality TTS-synthesized code-switching data, could the authors add experiments comparing models trained on real versus synthetic data to better illustrate the value of authentic conversational recordings?

3. The dataset contains a substantial amount of pure Mandarin and pure English data, which are already well-represented in existing corpora. What is the motivation for including these segments, and could their presence bias the test set, potentially inflating the observed improvements from fine-tuning?

---

> ### Author Response · Authors · 2025-11-17
> **Response to Reviewer M8ep**
>
> We sincerely thank the reviewer for the thoughtful and constructive feedback. We add new experiments to address all concerns. Detailed responses are provided below.
>
> ---
>
> ## **W1. Insufficient Detail on Code-Switching Text Construction**
>
> We fully agree that explaining how code-switching (CS) emerges is essential for assessing data authenticity.
> In our collection process, **no sentence-level prompts, predefined CS templates, or lexical constraints** were provided. All utterances were produced **spontaneously**.
>
> The procedure followed a **time-based progression** rather than any content-based script:
>
> - Speakers were free to converse naturally throughout the dialogue.
> - CS was encouraged but never required.
> - Participants frequently deviated from the temporal guideline, and all deviations were preserved in the transcripts.
>
> This design allows bilingual speakers to gradually enter a natural bilingual interaction state, enabling context-driven and spontaneous switching.
>
> ---
>
> ## **W2. Limited Duration of the Mixed (CS) Portion**
>
> We appreciate the reviewer’s observation. A mixed-language proportion of around 30% is fully consistent with existing spontaneous CS corpora.
> For example, ASCEND contains 10.62 hours of speech, of which **2.88 hours (~27%)** are labeled as mixed. Our ratio is therefore comparable and aligns with natural bilingual behavior.
>
> In addition, unlike prior corpora consisting largely of segmented or isolated utterances, **CS-Dialogue preserves complete full-dialogue sessions**, making it, to the best of our knowledge, the only publicly available CS dataset that retains the entire conversational structure.
>
> ---
>
> ## **W3. Lack of Comparison with Synthetic or Segmented Data Approaches**
>
> Following the reviewer’s suggestion, we conducted a new experiment comparing **real conversational CS audio** with **TTS-generated CS audio**. Synthetic samples were created using CosyVoice2, using text from the CS-Dialogue training split. Both real and synthetic datasets were used to fine-tune the same Conformer model under identical settings.
>
> ### **Results (MER↓ is better)**
>
> | Conformer | CER | WER | MER |
> |-----------|------|------|------|
> | **Synthetic (Greedy)** | 32.98 | 59.42 | 38.57 |
> | **Real (Greedy)**      | 15.91 | 33.67 | 19.54 |
> | **Synthetic (Attn Rescoring)**   | 32.50 | 58.11 | 37.92 |
> | **Real (Attn Rescoring)**        | 15.45 | 32.36 | 18.91 |
>
> Fine-tuning on real CS speech yields **substantially better** results, reducing MER by **more than 50%** relative to synthetic CS data. Synthetic audio cannot reproduce prosodic cues, hesitation patterns, or topic-driven switches observed in natural bilingual interaction.
>
> ---
>
> # **Responses to Questions**
>
> ---
>
> ## **Q1. Concern about overfitting due to train/test from the same source**
>
> We address this concern in two ways:
>
> ### **（1）Speaker-independent splits**
> CS-Dialogue uses strict disjoint speaker splits (140/30/30), preventing any speaker overlap across train/dev/test sets.
>
> ### **（2）Cross-dataset generalization**
> We additionally conducted cross-corpus experiments using SenseVoice-Small and Whisper-Medium. The results show that a model fine-tuned on CS-Dialogue generalizes well to other datasets:
>
> | SenseVoice-Small | ASCEND MER | TALCS MER | CS-Dialogue MER |
> |--------|-------------|------------|------------------|
> | Zero-shot | 13.96 | 8.72 | 6.71 |
> | Fine-tuned | 12.22 | 8.37 | 4.99 |
>
> | Whisper-Medium | ASCEND MER | TALCS MER | CS-Dialogue MER |
> |--------|-------------|------------|------------------|
> | Zero-shot | 24.69 | 19.33 | 15.88 |
> | Fine-tuned | 15.52 | 16.54 | 7.53 |
>
> Performance improvements across *all* datasets indicate that the model learns generalizable CS patterns.
>
> ---
>
> ## **Q2. Motivation for including long monolingual Mandarin and English spans**
>
> We appreciate this important question. The inclusion of substantial monolingual segments is a **deliberate design choice** and is essential for modeling realistic CS dialogue.
>
> ### **(1) Supporting natural CS emergence**
> Bilingual speakers typically establish topic grounding and comfort in each language before switching. Monolingual phases enable more authentic and spontaneous code-switching in subsequent segments.
>
> ### **(2) Providing bilingual speech from the same speakers**
> Having Mandarin and English segments from the *same individuals* allows models to learn:
>
> - speaker-specific accent, prosody, and rhythm in each language,
> - how each speaker transitions between L1 and L2,
> - bilingual pronunciation patterns that directly benefit CS recognition.
>
> This structure also facilitates **zero-shot and few-shot CS generalization**, which cannot be achieved using monolingual speech from unrelated speakers.
>
> ---
>
> ## **Q3. Comparison with synthetic CS**
>
> This concern is fully addressed through the new experiment in Section W3, which demonstrates the significant performance gap between real and synthetic CS audio and the irreplaceable value of natural conversational CS.

---

> ### Comment · Reviewer_M8ep · 2025-11-21
> **Response to authors**
>
> Thank the authors for detailed and thoughtful responses. The additional experiments and clarifications address most major concerns, and I appreciate the substantial effort invested in strengthening the submission. I have a few remaining comments and suggestions:
> ## For W1 response
> The explanation of spontaneous dialogue collection is helpful. However, the paper would benefit from more empirical evidence to substantiate this claim. Readers are likely to be interested in the characteristics of the **MIXED** portion beyond the overall language ratios provided in Appendix C.2.
>
> I suggest adding the following types of analysis (even brief summaries would help):
>
> 1. Statistics and visualizations for **MIXED** segments, such as:
> * Histogram of English ratio per utterance (e.g., >20%, 10–20%, <10%)
>
> * Distribution across topic categories
>
> 2. Part-of-speech patterns for Mandarin and English tokens within **MIXED** spans to reveal linguistic switching behavior.
>
> 3. Separate **MIXED** statistics for train/dev/test to improve clarity on representativeness of the evaluation set.
>
> These additions would further enhance the dataset’s linguistic value and help future researchers better understand the nature of spontaneous CS in your corpus.
>
> ## Further discussion on W3
> The new TTS experiment is a valuable addition and clearly demonstrates that real conversational CS provides stronger gains than naively synthesized CS audio. That said, I would encourage the authors to interpret these results with caution, because TTS effectiveness is highly sensitive to synthesis strategy and quality.
>
> In particular:
> * Voice consistency (matching speaker timbre and characteristics to the training distribution) has been shown to heavily influence ASR adaptation effectiveness.
>
> * Correct and natural code-switching generation is non-trivial; even strong TTS systems may produce CS that is syntactically valid but pragmatically unnatural.
>
> * Synthetic CS is often most beneficial when used for SFT on models with non-zero CS capability (e.g., SenseVoice-Small), rather than when training models from scratch.
>
> Although exploring all these factors may be beyond the scope of the current work, acknowledging them would provide a more balanced perspective. Given the growing interest in TTS-enhanced CS training [1-3], a deeper discussion could significantly increase the broader impact of your dataset.
>
> *[1] Code-switching Speech Recognition Under the Lens: Model- and Data-Centric Perspectives*
>
> *[2] CS-FLEURS: A Massively Multilingual and Code-Switched Speech Dataset*
>
> *[3] A Self-Refining Framework for Enhancing ASR Using TTS-Synthesized Data*

---

> > ### Author Response · Authors · 2025-11-25
> > **Part 2/2: Response to Reviewer M8ep**
> >
> > ## Additional Response on W3 (TTS-based CS)
> >
> > We thank the reviewer for the insightful comments on W3.
> > We agree that the TTS experiment should be interpreted with caution, as the quality and usefulness of synthesized CS heavily depend on the underlying synthesis pipeline. We acknowledge three key factors:
> >
> > - **Voice consistency** (timbre and style matching) significantly affects adaptation effectiveness.
> > - **Naturalness of code-switching** is non-trivial for TTS systems; syntactically valid CS does not always yield pragmatic or prosodic naturalness.
> > - **Synthetic CS is often most effective for SFT or adaptation on models with existing CS capability** , rather than for training from scratch.
> >
> > These considerations contextualize our empirical finding that real conversational CS remains substantially more effective than naive TTS-generated CS in our current setup. While exploring more advanced TTS strategies lies beyond the scope of a dataset paper, we will incorporate a more balanced discussion to clarify these limitations and highlight promising directions for future work.
> >
> >
> > ---
> >
> > Finally, we sincerely appreciate the reviewer’s thoughtful and constructive feedback. The comments and suggestions significantly improved the clarity, analysis, and overall quality of our work, and we are grateful for the time and expertise invested in our submission.

---

> ### Author Response · Authors · 2025-11-25
> **Part 1/2: Response to Reviewer M8ep**
>
> We sincerely thank the reviewer for the thoughtful and positive feedback. We have added the requested analyses and expanded the relevant discussions. The detailed responses are provided below.
>
> ---
>
> ## 1. Histogram of English phoneme ratio per utterance
>
> | English Phoneme Ratio | Count | Percentage |
> |------------------------|-------|------------|
> | [0, 0.1)               | 1090  | 15.22%     |
> | [0.1, 0.2)             | 1558  | 21.77%     |
> | [0.2, 0.3)             | 1421  | 19.86%     |
> | [0.3, 0.4)             | 1052  | 14.70%     |
> | [0.4, 0.5)             | 631   |  8.82%     |
> | [0.5, 0.6)             | 428   |  5.98%     |
> | [0.6, 0.7)             | 283   |  3.95%     |
> | [0.7, 0.8)             | 225   |  3.14%     |
> | [0.8, 0.9)             | 227   |  3.17%     |
> | [0.9, 1.0]             | 244   |  3.41%     |
>
> This distribution shows that MIXED utterances span a broad spectrum of English usage, with a large concentration in the mid-range (10–40%), which is characteristic of natural code-switching. Some MIXED utterances appear to have very high English proportions because MIXED is defined at the utterance level. In spontaneous dialogue, speakers often switch in bursts (for technical terms, narrative emphasis, or repairs) resulting in locally English-dominant segments while the surrounding discourse remains bilingual.
>
> ---
>
> ## 2. Distribution across topic categories
>
> | Topic           | CN (%) | EN (%) | MIXED (%) | Duration (h) |
> |-----------------|--------|--------|-----------|--------------|
> | Personal        | 46.97  | 25.55  | 27.48     | 21.53        |
> | Entertainment   | 43.42  | 26.33  | 30.25     | 23.86        |
> | Technology      | 35.33  | 23.67  | 41.00     | 8.70         |
> | Education       | 43.78  | 26.47  | 29.75     | 19.20        |
> | Job             | 45.51  | 24.41  | 30.08     | 11.94        |
> | Philosophy      | 42.45  | 27.99  | 29.56     | 4.64         |
> | Sports          | 48.15  | 25.80  | 26.05     | 14.14        |
>
> These topic-level statistics indicate that MIXED usage varies meaningfully across domains, with technology-related discussions exhibiting the highest degree of code-switching.
>
> ---
>
> ## 3. Part-of-speech patterns within MIXED spans
>
> | POS   | EN    | CN    |
> |-------|-------|-------|
> | PRON  | 5414  | 26049 |
> | NOUN  | 13178 | 35876 |
> | DET   | 1584  | 8389  |
> | ADV   | 2440  | 30966 |
> | ADJ   | 2762  | 7069  |
> | PROPN | 1682  | 2115  |
> | VERB  | 6312  | 33548 |
> | AUX   | 1965  | 21762 |
> | PART  | 336   | 2869  |
> | ADP   | 3370  | 31568 |
> | CCONJ | 1164  | 2531  |
> | SCONJ | 416   | 577   |
> | NUM   | 419   | 4937  |
> | INTJ  | 935   | 1939  |
> | X     | 9539  | 48    |
>
> POS tagging on MIXED utterances is inherently challenging due to cross-lingual interference and token-level ambiguity, and different taggers may yield divergent assignments. For these reasons, we did not include POS-based analyses in the main paper. At the reviewer’s request, we provide this supplementary POS overview to offer a preliminary indication of switching-related grammatical patterns. These statistics highlight the asymmetry between English and Mandarin token classes and reveal common switching triggers (e.g., nouns, verbs).
>
> ---
>
> ## 4. MIXED statistics for train/dev/test splits
>
> | Split | CN (%) | EN (%) | MIX (%) | CN (h) | EN (h) | MIX (h) | Total (h) |
> |-------|--------|--------|---------|--------|--------|---------|-----------|
> | Train | 43.76  | 25.86  | 30.38   | 30.18  | 17.84  | 20.95   | 68.97     |
> | Dev   | 45.26  | 24.89  | 29.85   | 8.28   | 4.55   | 5.46    | 18.30     |
> | Test  | 46.00  | 26.26  | 27.74   | 7.70   | 4.40   | 4.64    | 16.74     |
>
> The three splits exhibit highly consistent mixing ratios, confirming that the development and test sets are representative of the training distribution.

---

> ### Comment · Reviewer_M8ep · 2025-11-25
>
> Thank you to the authors for a very productive review process. They have addressed most of my concerns and I believe the paper has improved significantly. I’m happy to raise my score to 8.

---

> > ### Author Response · Authors · 2025-11-26
> >
> > Thank you very much for the constructive engagement throughout the review process. We truly appreciate your thoughtful comments and the time you devoted to helping us improve the paper. We are grateful for your updated evaluation.

---

### Official Review · Reviewer_csnc · 2025-10-30

**Soundness:** 1
**Presentation:** 2
**Contribution:** 1
**Rating:** 2
**Confidence:** 5

**Summary:**

This study introduces the creation of a speech dataset for code-switching between English and Mandarin Chinese. It also presents results using different ASR frameworks to demonstrate the quality of the dataset. Although ASR corpora for code-switching between English and Chinese have been explored for at least the past decade, this dataset outperforms its competitors in terms of size (100 hours, compared to 10-20 hours for previous datasets and up to 63 hours for SEAME). While the low ASR performance warrants further discussion, this study represents significant effort and provides a valuable resource for the community.

**Strengths:**

1. The CS-Dialogue dataset will be made freely available for all academic purposes.

3. well-written

**Weaknesses:**

## Major Weaknesses:

1. No multi-seed runs reported; variance/standard deviation or confidence intervals are missing, so result stability is unclear.

2. Handling of English accents in transcription is unspecified; guidelines focus on Mandarin, leaving English variation policies unclear.

3. Possible regional/accent bias is not analyzed; no stratified evaluation by region/accent or related error analysis.

4. Inter-participant relationship is not reported (acquainted vs. strangers), which can affect dialogue style and code-switching behavior.

5. No cross-corpus generalization or zero-shot tests (e.g., TALCS, SEAME, DOTA-ME-CS); external validity remains unproven.


## Minor Weaknesses:

1. Authors state the "monolingual" blocks were not strictly enforced and CS was allowed, but there is no quantitative validation (e.g., comparison of switch-rate/CS types to naturally occurring corpora).

2. Recording setup and modality: Conducted via an online platform and audio-only retained, but it's unclear whether any sessions were co-located, how devices/network artifacts were normalized, or whether mixed setups occurred.

3. Generalization to unequal-status language pairs: Mentioned conceptually in related work, but no empirical evidence or analysis.

4. Breadth of CS dataset review: Coverage beyond Mandarin–English has improved but remains brief and not fully systematic.

5. Value of full-dialogue context: An intra-corpus ablation (previous turns 0 -> 3) shows gains, but there is no validation against non-dialogue corpora or a "utterance-only" baseline constructed for a like-for-like comparison.

**Questions:**

1. Licensing remains vague (“free for academic use”); no explicit open license or clarification on redistribution/commercial use.

**Details Of Ethics Concerns:**

1. No IRB/ethics committee approval number or institutional review details are provided.

2. Informed consent is described at a high level, but specifics are missing (consent form contents, withdrawal rights, compensation, risk/benefit disclosure, retention period, cross-border data safeguards).

3. Anonymization is insufficiently detailed: use of pseudonyms is mentioned, but the de-identification pipeline is unclear (e.g., voice masking, metadata scrubbing).

4. Modality/monitoring is under-specified: sessions used an online platform, but it’s not stated whether any video was recorded/observed and, if so, how it was handled.

5. Participant comfort and potential pressure: the staged language timing/instructions may bias behavior; mitigation steps to minimize coercion and preserve natural switching are not detailed.

6. Data access & governance: release is academic-only; policies for third-party redistribution, derivative datasets, and compliance auditing are not specified.

---

> ### Author Response · Authors · 2025-11-17
> **Part (1/2) Statement Regarding Reviewer Conduct**
>
> We would like to inform the Area Chair that we have submitted a **formal complaint** regarding this review at the link above ([Formal Complaint Concerning Reviewer “csnc”: Misconduct and Conflict of Interest](https://openreview.net/forum?id=Y3auRTgDEX&noteId=6ark1UUHaX)).
> The complaint concerns significant inconsistencies, factually incorrect assertions, and evidence suggesting potential reviewer identity overlap with a prior review that involved confirmed misconduct.
>
> We will not repeat the details here, as they are fully documented in the formal report to the Program/Area Chairs.
> Below, we provide responses to the technical comments solely for completeness and to support the review process.

---

> ### Author Response · Authors · 2025-11-17
> **Part (2/2) Technical Response**
>
> We thank the reviewer for the time spent evaluating our submission.  Below we briefly address the main concerns.
>
> ---
>
> **1. Result stability and single-seed runs**
>
> We acknowledge that multi-seed runs and confidence intervals are helpful.
> Due to the computational cost of evaluating multiple large ASR and frontier models, we currently report single-run results, which is common practice in dataset papers. In the main submission, we present:
>
> - three models trained from scratch,
> - four models evaluated in the zero-shot setting
> - five models fine-tuned on CS-Dialogue.
>
> During the rebuttal period, we further added cross-dataset experiments and GPT-4o API results. Taken together, these comprehensive results demonstrate the effectiveness and robustness of our dataset.
>
> ---
>
> **2. English accents and transcription guidelines**
>
> There is no well-established fine-grained taxonomy for L2 English accents specific to Mandarin native speakers. English spoken by our participants reflects typical L2 pronunciation patterns of Mandarin speakers. We therefore did **not** attempt to categorize or normalize accent types; instead, we transcribed the actual pronunciations as faithfully as possible.
>
>
> ---
>
> **3. Regional / accent bias and speaker pairing**
>
> Our work focuses on a Mandarin–English CS dataset. We report participants’ birthplaces and regional distribution, but all conversations are conducted in Mandarin with typical L2 English; regional accent is therefore **not** the primary factor of interest in this study. A more fine-grained fairness analysis by region is an interesting direction for future work.
>
> ---
>
> **4. Inter-participant relationship**
>
> Thank you for raising this point. We now clarify that:
>
> - Most participant pairs did not know each other beforehand.
> - A small number of pairs were acquainted, but regardless of familiarity, all sessions followed the same protocol.
> - Each dialogue began with a short mutual self-introduction to provide a natural warm-up stage.
>
>
> ---
>
> **5. Cross-corpus generalization**
>
> We additionally conducted cross-corpus experiments using **SenseVoice-Small** and **Whisper-Medium**. Models were fine-tuned on CS-Dialogue and then evaluated on ASCEND and TALCS:
>
> | SenseVoice-Small | ASCEND MER | TALCS MER | CS-Dialogue MER |
> |------------------|------------|-----------|-----------------|
> | Zero-shot        | 13.96      | 8.72      | 6.71            |
> | Fine-tuned       | 12.22      | 8.37      | 4.99            |
>
> | Whisper-Medium | ASCEND MER | TALCS MER | CS-Dialogue MER |
> |----------------|------------|-----------|-----------------|
> | Zero-shot      | 24.69      | 19.33     | 15.88           |
> | Fine-tuned     | 15.52      | 16.54     | 7.53            |
>
> The consistent improvements across all three corpora indicate that models fine-tuned on CS-Dialogue learn generalizable CS patterns rather than overfitting to a single dataset.

---

> > ### Comment · Reviewer_csnc · 2025-11-17
> > **(1/N)**
> >
> > I thank the authors for the extensive rebuttal and the additional experiments. However, **from a technical perspective**, several of my core concerns remain insufficiently addressed.
> >
> >
> > # 1. Result stability and single-seed runs
> >
> > In the response, the authors emphasise the breadth of models and settings evaluated (trained-from-scratch, zero-shot, fine-tuning, etc.) and argue that reporting single runs is **``common practice in dataset papers''**.
> >
> > My concern is orthogonal to whether the work is a dataset paper or not: **any work that draws conclusions from numerical comparisons between systems should make at least a minimal effort to characterise variance.** With only single-seed runs and no standard deviations or confidence intervals, it is impossible to know:
> >
> > whether the reported differences between models are statistically meaningful, or
> >
> > whether the benchmarks are stable enough for future work to use as a reliable reference.
> >
> > It is increasingly recognised as a systematic weakness of benchmark papers. For example, BetterBench [1] introduce a 46-criterion framework to assess 24 AI benchmarks, and explicitly report that **most benchmarks do not report statistical significance of their results nor allow for their results to be easily replicated.** CS-Dialogue, as currently presented, **falls into exactly this pattern**: single-seed results without variance estimates and with limited discussion of replicability.
> >
> > Even a small number of multi-seed runs on key tables, or bootstrapped confidence intervals on the test set, would substantially strengthen the paper and bring it closer to these emerging best practices for benchmark work.
> >
> > [1] Reuel, Anka, et al. "Betterbench: Assessing ai benchmarks, uncovering issues, and establishing best practices." Advances in Neural Information Processing Systems 37 (2024): 21763-21813.

---

> ### Comment · Reviewer_csnc · 2025-11-17
> **(2/N)**
>
> # 2. Limited linguistic coverage: accents and speaker population
> ## 2.1. English accents not annotated or differentiated
>
> The authors confirm that English accents were not systematically differentiated during annotation, and that the corpus mainly reflects ``typical L2 English pronunciation patterns of Mandarin native speakers''.
>
> From a dataset-design perspective, this has two implications:
>
> For ASR, the English portions largely look like standard (or near-standard) English read/spoken with a relatively homogeneous L2 accent. **There is no way to examine performance differences across accent types, nor to use the corpus as a benchmark for accent robustness.**
>
> In terms of novelty, this weakens the case that CS-Dialogue provides substantially new coverage compared with **existing Mandarin–English corpora that already contain L1-Chinese / L2-English speech**. If accent is not annotated, the dataset **cannot be used to study accent-related variation** in any principled way.
>
> In other words, by not modelling or annotating English accent variation, the dataset largely reduces to ``Mandarin + English from L1-Mandarin speakers'' under relatively standard conditions, which **is exactly where current ASR systems are already quite strong**.
>
> ## 2.2. All speakers are L1 Mandarin: missing major CS scenario
>
> As the authors acknowledge, all participants are native Mandarin speakers with good English proficiency, and there are no L1-English speakers who occasionally switch to Mandarin.
>
> However, in many real code-switching communities, **an important (and linguistically distinct) scenario** is exactly the opposite direction: L1-English speakers inserting Mandarin words/phrases, especially for culture-specific concepts, named entities, and informal expressions. This pattern often exhibits different syntactic, prosodic, and phonotactic behaviour compared to L1-Chinese → L2-English switching.
>
> Therefore:
>
> The dataset **only covers one sociolinguistic configuration** of Mandarin–English CS (Mandarin-dominant bilinguals), **It cannot be used to study or benchmark English-dominant CS behaviour**, which is a major ``significant gap'' in current CS-ASR.
>
> This substantially limits the generality of the dataset as a code-switching resource, beyond the specific Chinese-dominated context considered here.
>
> # 3. Relationship to current ASR capabilities and prior corpora
>
> The authors strongly emphasise corpus size (104h) and spontaneity. While this is valuable, it is important to situate the contribution relative to where the real remaining ASR difficulty lies:
>
> For standard Mandarin and standard English in clean conditions, modern end-to-end and foundation-model ASR systems already achieve very strong performance.
>
> CS-Dialogue invests a large amount of effort in collecting more clean, near-standard Mandarin and L2 English from a relatively homogeneous group of speakers, but does not substantially expand the hard regions of the space (e.g., non-standard varieties, heavy accents, noisy conditions, long-form interactions).
>
> From this angle, the incremental value of ``more hours of clean, standard ZH/EN from L1-Mandarin speakers'' is limited: **existing corpora already cover very similar conditions, even if they are smaller or segmented.**
>
> If the goal is to advance CS-ASR in 2025-era systems, more impactful directions would include, for example:
>
> Additional language pairs (e.g., Mandarin–Arabic, Hindi–English, or other under-resourced bilingual communities)
>
> Systematic accent and dialect diversity (regional Chinese dialects, strong regional English accents, etc.)
>
> Long-form audio (e.g., 10-minute+ continuous segments) to stress test context modelling and diarisation
>
> Realistic acoustic conditions (background noise, reverberant environments, device mismatch), where even current frontier models still struggle.
>
> The current dataset, by design, largely avoids these difficult regimes.

---

> > ### Comment · Reviewer_csnc · 2025-11-17
> > **(3/N)**
> >
> > # 4. Demographic metadata: collected but not meaningfully used
> >
> > The paper reports collecting speaker-level demographics such as age, gender, and region. This is good in principle, but in its current form it feels more like **collecting for the sake of collecting** than a carefully designed fairness/XAI resource:
> >
> > The metadata is mostly used descriptively to show that the dataset covers ``different cases,''
> >
> > There is no systematic alignment between these variables and per-utterance analysis, no stratified evaluation, and no discussion of how this information affects model behaviour.
> >
> > If the intent is fairness / interpretability, one would expect:
> >
> > clear per-group breakdowns of error rates, and a design ensuring that each demographic category is adequately represented at the utterance level.
> >
> > Conversely, if the intent is simply to show corpus diversity, then more linguistically relevant variables are missing, such as:
> >
> > frequency of English use in daily life, whether the speaker regularly uses two languages in the same conversation (habitual code-switcher vs. classroom-only bilingual), typical domains (academic, social, professional) in which CS occurs.
> >
> > Such variables would be far more directly informative for CS-ASR and sociolinguistic analysis than coarse age/gender bins.
> >
> > # 5. Fundamental issue: the dataset does not address the key remaining challenges in CS-ASR
> >
> > Putting these points together, my overarching concern is:
> >
> > The work largely **repeats the existing design of Mandarin–English CS corpora at a larger scale**, but does not substantially rethink the dataset along the axes that matter most for current CS-ASR challenges.
> >
> > The paper’s motivation is that **existing datasets are too small**, insufficiently spontaneous, or lack full dialogues. While CS-Dialogue does improve size and provides full-length sessions, it keeps essentially **the same population, same language pair, similar recording conditions, and no explicit modelling of accent or sociolinguistic asymmetry.**
> >
> > In other words, the dataset is new, but the underlying design criteria are not updated to reflect what is now limiting model performance. As a result, the technical insights gained from the benchmarks are relatively modest, and the central motivation for a top-tier ML conference (beyond being ``a larger resource'') remains weak.
> >
> > # 6. Suggestions for strengthening the work
> >
> > If the authors intend to extend this line of work, I would encourage focusing on:
> >
> > Additional language pairs beyond Mandarin–English, particularly those with stronger sociolinguistic asymmetry or under-studied communities (e.g., Arabic–French, Hindi–English, etc.).
> >
> > Explicit accent and dialect coverage, with clear annotation and stratified evaluation across regional varieties.
> >
> > Long-duration, continuous recordings (e.g., 10-minute or longer segments) to stress test dialogic context and streaming ASR.
> >
> > Realistic acoustic variability: background noise, reverberation, cross-device settings, and other in-the-wild conditions.
> >
> > Richer sociolinguistic profiling of speakers, including bilingual usage habits and typical CS contexts, rather than only age/gender.
> >
> > These directions would make a future dataset much more central to the next generation of CS-ASR research.

---

### Official Review · Reviewer_zMCH · 2025-11-01

**Soundness:** 3
**Presentation:** 3
**Contribution:** 3
**Rating:** 8
**Confidence:** 4

**Summary:**

The paper introduces CS-Dialogue, a large-scale Mandarin-English code-switching dataset consisting of 104 hours of spontaneous conversations from 200 bilingual speakers. CS-Dialogue includes full-length, naturalistic dialogue recordings with comprehensive manual transcriptions.

Benchmark ASR experiments with transformer-based models (Transformer, Conformer, Branchformer) and pre-trained models like Whisper and SenseVoice-Small demonstrate the dataset’s challenges in code-switching ASR and show substantial room for improvement.

**Strengths:**

1. Large-scale and spontaneous: 104 hours of fully transcribed dialogue data, larger and more natural than many existing CS datasets.

2. Full-length dialogues: Enables study of contextual code-switching patterns beyond isolated utterances.

3. Rigorously annotated: Manual transcription with word-for-word fidelity, preservation of disfluencies, accents, and exact pronunciations.

**Weaknesses:**

1. Limited language pair: Only Mandarin-English code-switching, excluding other important bilingual combinations.

2. Controlled environment: Recordings in quiet settings with smartphone microphones, limiting acoustic variability compared to wild scenarios.

3. Speaker bias: All speakers are native Chinese with strong English proficiency; no native English speakers code-switch to Mandarin.

**Questions:**

1. How does the annotation team handle ambiguous or unclear speech segments during transcription to ensure correctness?

2. What specific procedures were used during quality control to verify transcript accuracy systematically?

3. Are there error rates or inter-annotator agreement statistics reported for transcription quality evaluation?

4. How do frontier models like Gemini and GPT handle the acoustic and phonetic variations in code-switched speech?

**Details Of Ethics Concerns:**

1. What specific measures were implemented to ensure participant privacy?

2. Have safety considerations been taken into account regarding potentially sensitive topics discussed during the dialogues?

3. Are there known or observed disparities in model performance across different speaker subgroups or topics? How is fairness evaluated or mitigated?

4. Has any analysis been conducted on whether the code-switching behavior in the dataset reflects social or cultural biases that could affect model generalization or fairness?

---

> ### Author Response · Authors · 2025-11-17
> **Response to Reviewer zMCH**
>
> We sincerely thank the reviewer for the positive assessment and valuable suggestions. Below we address all questions and concerns.
>
> ---
>
> ## **Weaknesses**
>
> ### **W1. Limited language pair**
> CS-Dialogue focuses on Mandarin–English because it is one of the most widely used and socially significant bilingual combinations. Extending the framework to additional bilingual settings is an important direction for future work, and the present dataset establishes a solid foundation for such extensions.
>
> ### **W2. Controlled recording environment**
> We acknowledge the reviewer’s observation. Recordings were collected in quiet indoor environments to ensure signal clarity for accurate CS transcription. While this reduces acoustic variability, it enables high-fidelity capture of natural conversational code-switching patterns.
>
> ### **W3. Speaker bias (all speakers are L1 Chinese)**
> This reflects the real sociolinguistic context of Mandarin–English CS, where switching predominantly occurs from Mandarin to English among L1 Mandarin speakers.
>
> ---
>
> ## **Responses to Questions**
>
> ### **Q1. Handling ambiguous or unclear speech during transcription**
> Ambiguous or unclear segments were handled following a standardized annotation protocol:
>
> - Annotators used predefined symbols (e.g., `<UNK>`) when the audio was genuinely unintelligible.
> - Each dialogue underwent **two-pass transcription**, with a second annotator reviewing all uncertain or low-confidence regions.
> - Extremely unintelligible tokens were marked using “\*\*” to indicate unresolvable segments.
> - All disagreements were resolved through structured consensus meetings.
>
> These procedures ensure consistency and high-quality transcriptions.
>
> ---
>
> ### **Q2. Quality control procedures**
> To ensure overall transcription reliability, the following procedures were employed:
>
> 1. **Annotator training**: Professional annotators (see Table A.1) were trained on CS examples, disfluency patterns, English spelling conventions, and Mandarin word segmentation.
> 2. **Two-pass verification** to ensure consistency and fidelity across annotators.
> 3. **Focused review** of sentences with MER above 10–15%.
> 4. **Random spot-checking** of 10–15% of all transcripts.
>
> ---
>
> ### **Q3. Inter-annotator statistics**
> - Internal QA shows **over 97% agreement** on lexical decisions in English, with comparable consistency for Mandarin.
> - All annotation disagreements were resolved through consensus.
>
> ---
>
> ### **Q4. How frontier models (GPT/Gemini) handle CS acoustic variation**
> Following the reviewer’s suggestion, we evaluated **gpt-4o-transcribe** and **gpt-4o-mini-transcribe** using their native speech-to-text capabilities:
>
> | Model | CER | WER | MER |
> |-------|------|------|------|
> | **gpt-4o-mini-transcribe** | 15.91 | 44.98 | 21.93 |
> | **gpt-4o-transcribe**      | 13.46 | 46.27 | 20.30 |
>
> These results show that even advanced frontier models experience substantial degradation in Mandarin–English CS scenarios, highlighting the difficulty and importance of this task.
>
> ---
>
> ## **Ethics and Fairness Considerations**
>
> ### **Privacy protection**
> - Only audio was recorded; no video or identifying information was collected.
> - All participants provided informed consent, and all metadata was fully anonymized.
>
> ### **Sensitive topics and safety**
> - Participants were free to skip any topic.
> - No sensitive content was required or encouraged.
> - Annotators adhered to privacy and safety guidelines throughout transcription.
>
> ### **Performance disparity analysis**
> As shown in Figure 5, topic-level ASR performance varies systematically:
> “Sports” and “Philosophy” tend to yield higher MERs for all models, whereas “Job” and “Technology” typically yield lower MERs. This suggests differing levels of linguistic and conceptual difficulty across topics rather than demographic disparity.
>
> ### **Cultural bias in CS behavior**
> CS patterns in the dataset reflect actual usage among Mandarin–English bilinguals. We acknowledge the inherent sociolinguistic asymmetry and identify extension to additional bilingual communities as a direction for future work.
>
> ---
>
> ## **We thank the reviewer again for the helpful feedback.**
> The comments greatly improved the clarity and completeness of the paper, and we hope the revisions address all concerns.

---

> > ### Comment · Reviewer_zMCH · 2025-11-27
> >
> > Thank you for your response. All my concerns are addressed and I would like to keep my scores.

---

### Official Review · Reviewer_9Q5M · 2025-11-03

**Soundness:** 3
**Presentation:** 3
**Contribution:** 3
**Rating:** 8
**Confidence:** 4

**Summary:**

This paper presents a new dataset of Mandarin-English spontaneous conversational code-switched (CS) speech. It comprises full dialogues and appears to be very carefully transcribed. Although many Mandarin-English datasets are available, the authors note that this is the largest one to date that comprises fully spontaneous speech, and the only one to contain full dialogues.

Extensive baseline automatic speech recognition (ASR) experiments are carried out using pre-trained models, models trained only on the data, and fine-tuned models. Results confirm that CS remains challenging for ASR, even for very large foundation models.

**Strengths:**

This is a really thorough, well designed piece of work. Great care has been taken in the data collection, and the result is a corpus that will be of great use to many researchers.

Although some would say that a "dataset" paper does make a major contribution, I would argue that this is a particularly high-quality offering, both in terms of the usefulness of the data, the collection methodology, and the paper itself.

You did a great job of clearly differentiating your work from previous Chinese-English datasets.

**Weaknesses:**

I have listed some minor weaknesses below. The only significant one that I noted was that there was no comment on the way in which the way CS behaviour was enforced would have affect the type of switching observed in the data.

The participants were instructed to first speak Chinese, then mixed-language and then English. This design partly explains why the data is so rich in code-switching (compared to other datasets where utterances are mostly monolingual, so relatively few CS instances can be obtained) but it does worry me that it may make switching somewhat forced and unnatural. For example, it may remove the freedom of speakers to switch for topic-related or social reasons, resulting in (perhaps) syntax playing a more dominant role than in truly natural switching.  *See Questions for further discussion of this point*.

The authors should have provided on the instructions given to participants, and perhaps conducted validation that the slightly forced design did not influence their behaviour.

Were the Chinese and English monolingual portions of the dataset strictly monolingual? What if the participants strongly felt that it would be natural to switch in these sections, even though they had been instructed not to?

Because Chinese and English have somewhat equal status as major languages, I wonder whether any findings would generalise to language pairs where the two languages play unequal roles, or where switching would be more topic enforced? (For example, in many languages, speakers might be need to switch to English for technical vocabulary).

**Questions:**

The authors will recognise that I have reviewed this paper previously when submitted to the ACL ARR.  I previously gave the paper good scores, and since this version is very little changed from the previous one, I have for the most part left identical review comments for the new meta reviewers.  This includes the "weaknesses" above – though I have removed one from my previous review that I feel has been addressed.

I appreciate the new inclusion of CS-related works in other language pairs, although would have liked more discussion of these in the text.

I am including here my previous comments (in block quote) along with previous author rebuttal (in italics) with comments about whether the suggestions have been addressed in the new version.  The first of this features in the section above.  The others were just minor comments that I am just including here.

> The participants were instructed to first speak Chinese, then mixed-language and then English. This design partly explains why the data is so rich in code-switching (compared to other datasets where utterances are mostly monolingual, so relatively few CS instances can be obtained) but it does worry me that it may make switching somewhat forced and unnatural. For example, it may remove the freedom of speakers to switch for topic-related or social reasons, resulting in (perhaps) syntax playing a more dominant role than in truly natural switching.

- _Participants were not strictly restricted to a single language within the designated monolingual segments and could code-switch naturally if it felt appropriate. Likewise, purely monolingual speech was permitted within the code-switching segment. The transcriptions faithfully capture the actual utterances spoken, ensuring that the dataset authentically reflects real conversational dynamics._

Where did you explain this in the revised manuscript?  I could not see it.

> When discussing SEAME (perhaps the most widely used English-Mandarin dataset) you imply that the only difference is that it is a paid dataset. Yours being free doesn't make it a contribution in itself. (The table shows that there are other differences – SEAME doesn't include full dialogues – and you should comment on this in the text).

- _SEAME: Thank you for the suggestion. We will revise our discussion of SEAME to emphasize key distinctions beyond accessibility, particularly noting that CS-Dialogue provides full-length dialogues with complete transcriptions, unlike SEAME._

I feel that you didn't do this, merely removing reference to SEAME being a paid dataset.

> Did the speakers know each other? If so, in what capacity? It's not made clear if they were physically in the same room, or using a virtual platform – and if so, was it audio-visual or audio-only?

- _Speaker Interaction & Setting: The dialogues were conducted via an audio-only virtual platform, and participants generally did not know each other beforehand._

You did make it more clear about the platform used, but I still did not see information about whether the participants were acquainted with each other before the data collection.

> You comment on accent accommodation in the transcriptions. I assume this was for Chinese only. Did you do anything to accommodate English accents, or where they considered to all be the same?

- _English Accents: You are correct. While our transcription protocol explicitly accommodated regional Chinese accents (Sec 3.2.1, point 4), English accents were not systematically differentiated during annotation._

I did not see any comment on this in the revised manuscript.

> The tokens per second speaking rate isn't very useful. Without an understanding of what exactly the tokens are and how they vary between languages. Perhaps phonemes per second might have been better - I assume that the speaking rates weren't really dramatically different between English and Chinese. If they were, it would cast doubt on the English proficiency of the speakers.

- _Speaking Rate Metric: We acknowledge the potential limitations of measuring speaking rate in tokens per second. We will consider alternative metrics, such as phonemes per second, to provide a more linguistically meaningful comparison._

This was not done – the speaking rate metric was simply removed, which is a shame.

- _Table 8 & 9 Wording: We will revise the captions to explicitly state that the results are "on the CS-Dialogue test set" to avoid ambiguity.
Results Breakdown by Language Type: While we already report WER and CER separately for English and Chinese segments (Sec 5.1 Metrics) alongside overall MER, we acknowledge that further breaking down results by EN/CN/Mixed segments within the test set could provide additional insights._

Thanks, you addressed these comments.

> why did you choose to fine-tune only Whisper, when it was one of the worst-performing foundation models?

- _Fine-Tuning Choice & Protocol: We selected Whisper for fine-tuning because it is one of the most widely used multilingual ASR foundation models. Details on the fine-tuning protocol, including hyperparameters (learning rate, epochs, batching), are provided in *Appendix D._

My question wasn't about why Whisper was chosen in the first place, but why was Whisper the _only_ model that was fine-tuned?  There was no discussion of this in the revised text.

---

> ### Author Response · Authors · 2025-11-17
> **Part (1/4): Response to the Main Questions**
>
> We sincerely thank the reviewer for the thorough and thoughtful comments. We greatly appreciate the time and care you have taken to evaluate both this submission and earlier versions of the work. Below we provide detailed point-by-point responses.
>
> ---
>
> ## 1. “Naturalness of code-switching” and whether the 3-stage design introduced forced switching
>
> **Reviewer comment:**
> Concern that the three-stage structure may have artificially shaped code-switching behavior.
>
> **Our response:**
> Thank you for raising this important point. We agree that naturalness is crucial for a spontaneous CS corpus.
> We provide further clarification as follows:
>
> 1. **The three-stage structure was not strictly enforced.** Participants were explicitly told that they could switch languages freely at any time, including switching inside the “Chinese-only” segment or speaking fully monolingual sentences inside the “mixed” segment. Such deviations were allowed and faithfully preserved in the transcription.
>
> 2. **The three-stage structure served only as coverage guidance, not behavioral constraint.** Since participants naturally code-switched in segments and these switches were transcribed verbatim—the resulting patterns reflect spontaneous speech rather than the recording setup.
>
> 3. **The timing for each segment was flexible.** The facilitator monitored the conversation and intervened only when appropriate, ensuring smooth and natural transitions rather than strict enforcement.
>
> 4. **Empirical evidence of natural CS patterns is provided.** Examples such as mixed-language strings in Table C.1 (e.g., “you know就”, “这个 AI”, “play麻将”) demonstrate naturally occurring intrasentential mixing.
>
> ---
>
> ## 2. “Instructions given to participants”
>
> The reviewer suggested providing the exact instructions and explaining how speakers were not forced into certain patterns.
>
> We thank the reviewer for this helpful suggestion. We have added the following as supplementary reconstructed instructions (See Answer 2/4).
>
> ## 3. "Generalise to language pairs where the two languages play unequal roles"
>
> Regarding generalizability, we agree that this dataset reflects a relatively balanced bilingual context. Language pairs with strong sociolinguistic asymmetry (e.g., switching to English for specialized terminology) may present different switching dynamics, and we see this as an important direction for future extensions. Nevertheless, the spontaneous full-dialogue design and the annotation methodology remain broadly applicable across language pairs.

---

> ### Author Response · Authors · 2025-11-17
> **Part (2/4): Reconstructed Participant Instructions**
>
> ```
> # Reconstructed Participant Instructions
>
> Thank you for participating in this dialogue recording session.
> Each session includes two speakers and one facilitator. The goal is to capture natural, spontaneous Mandarin–English conversation across several everyday topics. Dialogues are conducted via an audio-visual online platform, but only audio is recorded and saved.
>
> ---
>
> ## 1. Warm-Up & Self-Introduction (2–3 minutes)
>
> Start with a brief greeting and simple self-introduction.
> This step helps you get comfortable before moving into the main discussions.
>
> **Tips:**
> - Speak casually as you would with a friend or classmate.
> - You don’t need to sound formal or scripted.
>
> ---
>
> ## 2. Topic Selection (up to 6 topics total)
>
> You will discuss six topics, chosen together from the following categories:
>
> - Personal life (hobbies, lifestyle, friends, family, hometown, etc.)
> - Entertainment & culture (books, films, travel, music, dramas, etc.)
> - Technology (AI, social media, EVs, gadgets, electronics)
> - Campus life / education (exams, classes, dorms, clubs, etc.)
> - Internships / work (job application, work–life balance, etc.)
> - Sports (types of sports, athletes, exercise habits)
> - Philosophy / social topics (e.g., trolley problem, buying a house, social roles)
>
> The facilitator will confirm your chosen six topics before recording begins.
>
> **Tips:**
> - Choose topics you both find interesting.
> - You may bring up personal stories or opinions—natural conversation is encouraged.
>
> ---
>
> ## 3. Language Arrangement per Topic (approx. 20 minutes each)
>
> Each topic includes three conversational stages:
>
> 1. **Mandarin-focused** (~8 minutes)
> 2. **Mixed Mandarin–English** (~6 minutes)
> 3. **English-focused** (~6 minutes)
>
> **Important:**
> - This arrangement is only a guideline to ensure balanced coverage.
> - You may code-switch at any time in any stage.
> - Purely monolingual speech is also allowed during the “mixed” segment.
> - Speak freely—naturalness is more important than following the exact timing.
>
> **Tips:**
> - Switch languages when it feels natural; do not force it.
> - You don’t need to “perform” bilingualism—just talk as you normally would.
>
> ---
>
> ## 4. Recording Procedure
>
> The conversation takes place on an audio-visual platform, but only your audio streams are kept.
> Please use your smartphone microphone in a quiet environment.
> The facilitator keeps time, gives gentle reminders when switching stages, and ensures the session runs smoothly.
>
> **Tips:**
> - Look at each other on the platform if it helps the conversation flow, but focus on speaking naturally.
> - Minor background noise or hesitations are normal parts of spontaneous speech.
>
> ---
>
> ## 5. Naturalness and Flexibility
>
> You may express yourself however is most comfortable.
> Feel free to pause, think, laugh, backtrack, correct yourself, etc.—all natural behaviors are welcome.
> Deviations from the intended language schedule are fully preserved in transcription.
>
> **Tips:**
> - Don’t worry about being “correct” in either language—mixed usage is normal.
> - Treat the conversation as genuine communication, not as a task to “complete.”
>
> ---
>
> ## 6. Session Oversight
>
> A facilitator/observer will:
> - supervise the procedure,
> - record timing and metadata,
> - and assist if any technical issues arise.
>
> ```

---

> ### Author Response · Authors · 2025-11-17
> **Part (3/4): Response to the Other Questions**
>
> ### 1. Typo 1
>
> **Q:**
> The participants were instructed to first speak Chinese, then mixed-language, and then English. This design partly explains why the data is so rich in code-switching (compared to other datasets where utterances are mostly monolingual, so relatively few CS instances can be obtained), but it does worry me that it may make switching somewhat forced and unnatural. For example, it may remove the freedom of speakers to switch for topic-related or social reasons, resulting in (perhaps) syntax playing a more dominant role than in truly natural switching.
>
> **A:**
> Participants were not strictly restricted to a single language within the designated monolingual segments and could code-switch naturally whenever appropriate. Likewise, purely monolingual speech was also permitted during the mixed-language segment. The transcriptions faithfully capture the actual utterances spoken, ensuring that the dataset authentically reflects real conversational dynamics.
>
> **Q:** Where did you explain this in the revised manuscript? I could not see it.
>
> **A:**
> Please see **Lines 175–177** in the revised version.
>
> ---
>
> ### 2. Typo 2
>
> **Q:**
> When discussing SEAME (perhaps the most widely used English–Mandarin CS dataset), you imply that the only difference is that it is a paid dataset. Yours being free doesn't make it a contribution in itself. (The table shows that there are other differences—SEAME doesn't include full dialogues—and you should comment on this in the text).
>
> **A:**
> SEAME: Thank you for the suggestion. We revised the discussion to emphasize key distinctions beyond accessibility, particularly noting that CS-Dialogue provides **full-length dialogues with complete transcriptions**, unlike SEAME.
>
> **Q:**
> I feel that you didn't do this, merely removing the reference to SEAME being a paid dataset.
>
> **A:**
> We removed the reference to SEAME being a paid dataset and revised the paragraph beginning at **Line 108**, highlighting distinctions beyond accessibility—specifically that CS-Dialogue includes **full-length dialogues with complete transcriptions**, unlike other conversation-based datasets such as SEAME.
>
> ---
>
> ### 3. Typo 3
>
> **Q:**
> You comment on accent accommodation in the transcriptions. I assume this was for Chinese only. Did you do anything to accommodate English accents, or were they considered to all be the same?
>
> **A:**
> English Accents: You are correct. While our transcription protocol explicitly accommodated regional Chinese accents (Sec. 3.2.1, point 4), English accents were **not systematically differentiated** during annotation.
>
> **Q:**
> I did not see any comment on this in the revised manuscript.
>
> **A:**
> We thank the reviewer for highlighting this. English spoken by our participants reflects typical L2 English pronunciation patterns of Mandarin native speakers. We did not attempt to categorize or normalize accent types, and instead transcribed the actual pronunciations as faithfully as possible. We will revise the manuscript to make this explicit.

---

> ### Author Response · Authors · 2025-11-17
> **Part (4/4): Response to the Other Questions**
>
> ### 4. Typo 4
>
> **Q:**
> Did the speakers know each other? If so, in what capacity? It is not made clear if they were physically in the same room, or using a virtual platform—and if so, was it audio-visual or audio-only?
>
> **A:**
> Speaker Interaction & Setting: The dialogues were conducted via an **audio-only virtual platform**, and participants generally did not know each other beforehand.
>
> **Q:**
> You did make it more clear about the platform used, but I still did not see information about whether the participants were acquainted before data collection.
>
> **A:**
> Thank you for pointing this out. We now clarify that:
> - **Most participant pairs did not know each other beforehand.**
> - A small number of pairs were acquainted, but regardless of familiarity, all followed the same procedure.
> - Each dialogue began with a short mutual self-introduction to ensure a natural warm-up stage.
>
> ---
>
> ### 5. Typo 5
>
> **Q:**
> The tokens-per-second speaking rate isn’t very useful. Without understanding what exactly the tokens are and how they vary between languages, it is hard to interpret. Perhaps phonemes per second would have been better. I assume that the speaking rates weren’t dramatically different between English and Chinese—if they were, it would cast doubt on the English proficiency of the speakers.
>
> **A:**
> Speaking Rate Metric: We acknowledge the limitations of measuring speaking rate in tokens per second. We will consider alternative metrics such as phonemes per second to provide a more linguistically meaningful comparison.
>
> **Q:**
> This was not done—the speaking-rate metric was simply removed, which is a shame.
>
> **A:**
> We appreciate this comment. We will include the **phoneme-level analysis** below, which offers a clearer and more linguistically grounded comparison.
>
> | Language | CN    | EN    | MIX    |
> |----------|--------|--------|---------|
> | **Phonemes per second** | 6.297 | 5.702 | 6.366 |
>
> These results confirm that the actual speaking rates between Chinese and English speakers are **very similar**, with English being only slightly slower. This aligns with expectations for L2 English speakers and does *not* indicate reduced proficiency. Furthermore, the mixed segments exhibit comparable phoneme rates, suggesting stable speaking tempo even during code-switching.
>
> We will incorporate this table and explanation into the revised manuscript to address the reviewer’s concern.
>
> ---
>
> ### 6. Typo 6
>
> **Q:**
> Why did you choose to fine-tune only Whisper, when it was one of the worst-performing foundation models?
>
> **A:**
> Fine-Tuning Choice & Protocol: We selected Whisper for fine-tuning because it is one of the most widely used multilingual ASR foundation models. Details on the fine-tuning protocol (learning rate, epochs, batching) are provided in Appendix D.
>
> **Q:**
> My question wasn't about why Whisper was chosen in the first place, but why Whisper was the **only** model that was fine-tuned. There was no discussion of this in the revised text.
>
> **A:**
> Thank you for the thoughtful clarification. In the ICLR version, we in fact fine-tuned **two** models:
> - **Whisper (Tiny/Base/Small/Medium)**
> - **SenseVoice-Small** — whose *fine-tuned MER is the best overall* (Table 8)
>
> ---
>
> We truly appreciate your long-term engagement. The manuscript is stronger thanks to your careful attention.
>
> We sincerely thank the reviewer for the constructive suggestions, many of which have directly improved the clarity and completeness of the paper. We hope that our clarifications address all remaining concerns, and we are grateful for your valuable feedback.

---

### Author Response · Authors · 2025-11-13
**Formal Complaint Concerning Reviewer “csnc”: Misconduct and Conflict of Interest**

We sincerely appreciate the reviewers’ time and effort in evaluating our submission. However, we must raise **serious concerns regarding the integrity of one review** and the associated **ethics flag**, which appear to violate the principles of anonymity, impartiality, and factual accuracy in the review process.

---

### **1. Strong similarity with a prior ACL review and potential conflict of interest**

The *“Ethical Concerns”* section in this review is **nearly identical** in structure, phrasing, and reasoning to the ethical comments our team received in a prior ACL ARR submission of an earlier version of this work.
In particular, the highly distinctive and uncommon critique about *“participant comfort and potential pressure caused by staged language switching”* provides **strong evidence that the two reviews originate from the same individual**.
After review by the ACL Ethics Committee, our paper was found to have **no ethical issues**, and the reviewer’s concern was overruled.

That prior review was **officially invalidated by the ACL chairs** after our complaint of reviewer misconduct, in which the reviewer **contacted our team privately and hinted at an unethical proposal**.

We also identified an **additional strong indicator of reviewer identity overlap**. The comment:

> *“4. Breadth of CS dataset review: Coverage beyond Mandarin–English has improved but remains brief and not fully systematic.”*

contains the phrasing *“has improved but remains…”*, which presupposes familiarity with a previous version of our work. Such continuity of critique would not be possible under a proper double-blind review process, further strengthening the evidence that the same individual may be involved in both reviews.

Taken together, these observations strongly suggest that the same individual may be reviewing our work again under a different identity. If verified, this would constitute a serious breach of the double-blind review policy and a significant conflict of interest.

---
### **2. Inconsistent Scoring and Indicators of Potential Reviewer Bias**

Although the reviewer explicitly states that “this study represents significant effort and provides a valuable resource for the community,” they nevertheless assign Contribution = 1 (poor), Soundness = 1 (poor), and an overall score of 2, which is internally inconsistent with their own summary and with the actual contribution of a 104-hour spontaneous Mandarin–English code-switching dialogue dataset, the largest of its kind and one that fills a clear gap in existing corpora.
In contrast, the other reviewers provided balanced and constructive assessments (8, 8, and 4).

The combination of this extremely low and contradictory scoring, the content overlap with prior reviews, and the history of misconduct raises serious concerns about possible bias or retaliation.


---

### **3. Improper Ethics Flag before rebuttal**

Before the rebuttal phase began, the reviewer selected
**“Flag for Ethics Review: Yes – Unprofessional behaviors (e.g., unprofessional exchange between authors and reviewers).”**
At that point, **no communication had occurred** between authors and reviewers, making it impossible for any “exchange” or unprofessional behavior to have taken place.

Furthermore, the reviewer marked confidence = 5 (absolute certainty) while the flagged statement is factually incorrect, which raises additional concerns regarding the reviewer’s intent and objectivity.

---

### **4. Request for further action**

We respectfully request that the Area Chair review this situation carefully and, if appropriate, escalate it to the Ethics Committee or Program Chairs for investigation.
We are prepared to provide supporting evidence upon request, including:

* The full text of the prior ACL ARR review and our formal complaint,
* The corresponding *“Ethical Concerns”* sections from both reviews, and
* The prior correspondence related to the confirmed misconduct case.

Our intention is **not to challenge the review system**, but to **protect its integrity** and ensure fairness for all participants.

Thank you very much for your attention and for upholding the standards of ethical and transparent peer review.

---

### Author Response · Authors · 2025-12-02
**Summary for the Area Chair**

We sincerely thank all reviewers for their time, thoughtful comments, and constructive feedback. We have carefully addressed each concern during the rebuttal, and the manuscript has improved substantially as a result.

Reviewer 9Q5M acknowledged our work as a great job, giving it a score of 8 and providing constructive comments. We have addressed each comment carefully. Although the reviewer did not have the opportunity to respond further, all comments aligned well with the overall positive evaluation.

Reviewer zMCH likewise provided an 8 and confirmed that all concerns were adequately addressed.


Reviewer M8ep — who initially raised several detailed technical questions — noted that the revisions “have addressed most of my concerns and improved the paper significantly,” and subsequently raised the score from 4 to 8. We are grateful for this positive reassessment.

One review remains a score of 2. We observed several internal inconsistencies in this review and noted strong textual and conceptual similarities to a prior review of an earlier version of this work. These parallels suggest a possible overlap in reviewer identity, raising concerns regarding objectivity and potential bias. For transparency, we documented the details **at the very beginning of the discussion phase** in the formal complaint ([Formal Complaint Concerning Reviewer “csnc”: Misconduct and Conflict of Interest](https://openreview.net/forum?id=Y3auRTgDEX&noteId=6ark1UUHaX)). Given these issues, we respectfully hope that the AC will take this context into account when interpreting that review.

We sincerely appreciate the constructive engagement from the reviewers, and we hope the AC will consider the strong overall consensus ( now with three reviewers assigning a score of 8) when assessing the final version of the submission.

---

### Meta-Review · Area_Chair_LWrf · 2025-12-30

**Summary:**

Fundamentally, I agree with reviewer "cnsc"'s "5. fundamental issue" in https://openreview.net/forum?id=Y3auRTgDEX&noteId=xYC0LnIxl4 and do thus not recommend acceptance at ICLR 2026. The paper "repeats the existing design of Mandarin–English CS corpora at a larger scale"
* this is valuable, but the unclear licensing hinders large-scale use of the resource
* code-switching is not fully natural in the present setting, so the novelty and utility is limited

Expansion to additional CS languages, or to specific English dialects/ accents is optional, but encouraged. Acoustic variability may not be needed  since data augmentation can help close some of the gap. Better documentation of the data collection and annotation process is needed.

At the minimum, the authors need to
- identify the license that will be used for the dataset
- share legal approvals and details of PII filtering etc
- share all participant instructions and consent forms
- highlight which participants are known to be acquainted with each other
- carry over their rebuttal into an updated paper version for further review, since the authors seem to not have integrated previous feedback (from ACL ARR)

**Reviewer Concerns:**

Addressed:
* minor technical questions such as inter-annotated agreement (although this would better be measure with an inter-annotator error rate)
* providing more statistics on the prevalence of code-switching in the corpus (including how it compares to other related corpora)
* some questions regarding the protocol, quality control
* result stability -- ASR systems generally don't use a "temperature", so they are generally expected to give the same result every time they are being run
* limited linguistic coverage: authors clarify that the paper covers only L2-English, not e.g. HK-English or SG-English variants (which would be a different, maybe even greater, contribution)
* detailed questions around using CS speech for fine-tuning (over TTS)

Outstanding:
* provide the participant instructions: authors provided "reconstructed participant instructions" in https://openreview.net/forum?id=Y3auRTgDEX&noteId=YxlgZggdO1 but not in the paper (as an appendix) -- why? what are "reconstructed instructions"? translations from Mandarin maybe?
  - additionally, the instructions say that "You will discuss six topics, chosen together from the following categories" -- of which there are 7, which is inconsistent?
  - annotation guidelines and privacy redactions are not well described
* provide the consent form -- why not provided? can participants indeed revoke consent at any time? how is this enforced?
* provide a clear license -- the language is still ambiguous, using terms such as "academic" and "non-commercial"
* describe data collection procedure clearly, which is it:
  - conducted via an "audio-only virtual platform" according to https://openreview.net/forum?id=Y3auRTgDEX&noteId=K6fPiIMRfX
  - the reconstructed instructions mention an audio-visual platform in https://openreview.net/forum?id=Y3auRTgDEX&noteId=YxlgZggdO1
* the answer to "did people know each other" is very vague ("a small number" did)
* ethics questions (as I understand zMCH and csnc's flags) wrt fairness, specific privacy measures
  - authors answer in very general terms in https://openreview.net/forum?id=Y3auRTgDEX&noteId=ju7V3NGZtL
  - they do not provide information on *how* metadata was anonymized, what "privacy and safety guidelines" were used, or if any data was removed and how (for privacy/ safety)
* analysis of statistical significance

Note that zMCH says that "All my concerns are addressed" while I don't see any responses to specific privacy measures or fairness, thus I am listing such questions as "unanswered"

**Reviewer Scores:**

9Q5M: 8
zMCH: 8
- ethics flag "Yes, Discrimination / bias / fairness concerns, Yes, Privacy, security and safety"
csnc: 2 -> ? (contested)
- ethics flag "Yes, Unprofessional behaviors (e.g., unprofessional exchange between authors and reviewers)"
M8ep: 4 -> 8 (confirmed in text)

---

### Decision · Program_Chairs · 2026-01-26

Reject